# The liprin-α/RIM complex regulates the dynamic assembly of presynaptic active zones via liquid–liquid phase separation

Gaowei Jin[1,2,3], Joaquín Campos[4], Yang Liu[1,2,3], Berta Marcó de la Cruz[5,6], Shujing Zhang[2,3], Mingfu Liang[2,3], Kaiyue Li[2,3], Xingqiao Xie[1,2,3], Fredrik H. Sterky[5,6,7]*, Claudio Acuna[4]*, Zhiyi Wei[1,2,3,8]*

**1** Shenzhen Key Laboratory of Biomolecular Assembling and Regulation, Southern University of Science and Technology, Shenzhen, China, **2** Brain Research Center, Southern University of Science and Technology, Shenzhen, China, **3** School of Life Sciences, Southern University of Science and Technology, Shenzhen, China, **4** Chica and Heinz Schaller Foundation, Institute of Anatomy and Cell Biology, University of Heidelberg, Heidelberg, Germany, **5** Department of Laboratory Medicine, Institute for Biomedicine, Sahlgrenska Academy, University of Gothenburg, Gothenburg, Sweden, **6** Wallenberg Centre for Molecular and Translational Medicine, University of Gothenburg, Gothenburg, Sweden, **7** Department of Clinical Chemistry, Sahlgrenska University Hospital, Gothenburg, Sweden, **8** Institute for Biological Electron Microscopy, Southern University of Science and Technology, Shenzhen, China

☯ These authors contributed equally to this work.
* fredrik.sterky@gu.se (FHS); acuna@uni-heidelberg.de (CA); weizy@sustech.edu.cn (ZW)

## Abstract

Presynaptic scaffold proteins, including liprin-α, RIM, and ELKS, are pivotal to the assembly of the active zone and regulating the coupling of calcium signals and neurotransmitter release, yet the underlying mechanism remains poorly understood. Here, we determined the crystal structure of the liprin-α2/RIM1 complex, revealing a multifaceted intermolecular interaction that drives the liprin-α/RIM assembly. Neurodevelopmental disease-associated mutations block the formation of the complex. Disrupting this interaction in cultured human neurons impairs synaptic transmission and reduces the readily releasable pool of synaptic vesicles. Super-resolution imaging analysis supports a role for liprin-α in recruiting RIM1 to the active zone, presumably by promoting the liquid–liquid phase separation (LLPS) of RIM1. Strikingly, the liprin-α/RIM interaction modulates the competitive distribution of ELKS1 and voltage-gated $Ca^{2+}$ channels (VGCCs) in RIM1 condensates. Disrupting the liprin-α/RIM interaction significantly decreased VGCC accumulation in the condensed phase and rendered release more sensitive to the slow calcium buffer EGTA, suggesting an increased physical distance between VGCC and vesicular calcium sensors. Together, our findings provide a plausible mechanism of the liprin-α/RIM complex in regulating the coupling of calcium channels and primed synaptic vesicles via LLPS for efficient synaptic transmission and uncover the pathological implication of liprin-α mutations in neurodevelopmental disorders.

**Data availability statement:** All relevant data are within the paper and its Supporting information files. The structure factors and atomic model of the liprin-α2_CC2N/RIM1_C2B complex have been deposited in the Protein Data Bank (PDB) with accession code 8Z22.

**Funding:** This work was supported by Key-Area Research and Development Program of Guangdong Province (Grant No. 2023B0303010001 to Z.W.), Shenzhen Science and Technology Program (RCJC20210609104333007 to Z.W.), the National Natural Science Foundation of China (32471250 to Z.W. and 32371009 to X.X.), Shenzhen-Hong Kong Institute of Brain Science, Shenzhen Fundamental Research Institutions (2023SHIBS0002 to Z.W.), Shenzhen Key Laboratory of Biomolecular Assembling and Regulation (ZDSYS20220402111000001 to Z.W.), the Chica and Heinz Schaller Foundation (C.A.), Brain and Behavior Research Foundation (C.A.), the Deutsche Forschungsgemeinschaft (DFG SFB1158-SO2 to C.A.), the Fritz Thyssen Foundation (10.21.0.019MN to C.A.), the Swedish Research Council (2022-00817 to F.H.S.) and the DAAD/ANID fellowship (57451854/62180003 to J.C.). F.H.S. is supported by the Knut and Alice Wallenberg Foundation via the Wallenberg Centre for Molecular and Translational Medicine at the University of Gothenburg. The funders had no role in study design, data collection and analysis, decision to publish, or preparation of the manuscript.

**Competing interests:** The authors have declared that no competing interests exist.

**Abbreviations:** aSEC, analytical size-exclusion chromatography; ITC, isothermal titration calorimetry; FRAP, fluorescence recovery after photobleaching; HEK, human embryonic kidney; hESCs, human embryonic stem cells; LLPS, liquid–liquid phase separation; MALS, multi-angle light scattering; mEPSCs, miniature excitatory postsynaptic currents; PEI, polyethylenimine; PIP$_2$, phosphatidylinositol 4,5-bisphosphate; RBP, RIM-BP; ROIs, regions of interest; RRP, readily releasable pool; Trx, thioredoxin; VGCC, voltage-gated Ca$^{2+}$ channel.

## Introduction

Synaptic transmission is the cornerstone of brain functions, representing the fundamental process through which neurons communicate. Triggered by an action potential, neurotransmitter release occurs at a specialized region in the presynaptic terminal, known as the "active zone" [1–3]. This dynamic region arises from the orchestrated assembly of a diverse array of proteins, forming an electron-dense structure attached to the plasma membrane that governs synaptic vesicle exocytosis. Five conserved scaffold proteins have emerged as core components in active zone assembly, namely liprin-α, RIM, RIM-BP (RBP), ELKS, and Munc13 [1]. However, the assembly mechanism remains poorly understood, mainly due to the limited understanding of the complex interactions among these proteins.

Among these core scaffolds, liprin-α has garnered increasing attention for its evolutionary conserved roles in active zone formation and function. The liprin-α family contains four members (liprin-α1/2/3/4) in mammals and one member each in *Caenorhabditis elegans* and *Drosophila* [4–8]. Genetic studies in invertebrates reveal that dysfunction of liprin-α orthologs leads to altered active zone morphology and diminished synaptic vesicle accumulation [5,6,9]. Although the depletion of two neuron-specific isoforms, liprin-α2 and α3, mildly disrupts active zone ultrastructure and vesicle priming in mice [10], knocking out all four liprin-α genes in human neurons blocks the recruitment of active zone components and synaptic vesicle [11], demonstrating the indispensable role of liprin-α proteins in mammalian presynaptic structure and function. Liprin-α organizes the active zone by interacting with various presynaptic proteins [6,10,12–16] (Fig 1A). Its C-terminal SAM domains associate with presynaptic adhesion molecules, such as LAR-type receptor protein tyrosine phosphatases and neurexins through forming the liprin-α/CASK/neurexin tripartite complex [11,17–19]. On the other hand, liprin-α employs N-terminal coiled coils to recruit other presynaptic scaffolds, including RIM and ELKS, to synaptic adhesion sites for active zone formation and function [15,20–23].

RIM binds to the other four core scaffold proteins and Ca$^{2+}$ channels [20,24–27], contributing to active zone assembly. The RIM family in vertebrates has four members [28]: RIM1 and RIM2 are multidomain-containing proteins with two C-terminal C2 domains (C2A and C2B; Fig 1A) involved in neurotransmitter release through phospholipid binding [29], whereas RIM3 and RIM4 only contain the C2B domain. RIM1 and RIM2 were identified to interact through their C2B domains with the coiled-coil region of liprin-α [20]. Although the recognized significance of both liprin-α and RIM in active zone formation and function, a molecular understanding of their interaction and its functional consequences remain unknown. Discovery of RIM/RBP condensation indicates the involvement of liquid–liquid phase separation (LLPS) in coupling synaptic vesicles and voltage-gated Ca$^{2+}$ channels (VGCCs) [30,31]. In agreement with this, liprin-α co-phase separates with ELKS, to support active zone formation [32]. These findings raise intriguing questions about how the liprin-α/RIM interaction may regulate these sophisticated protein assemblies.

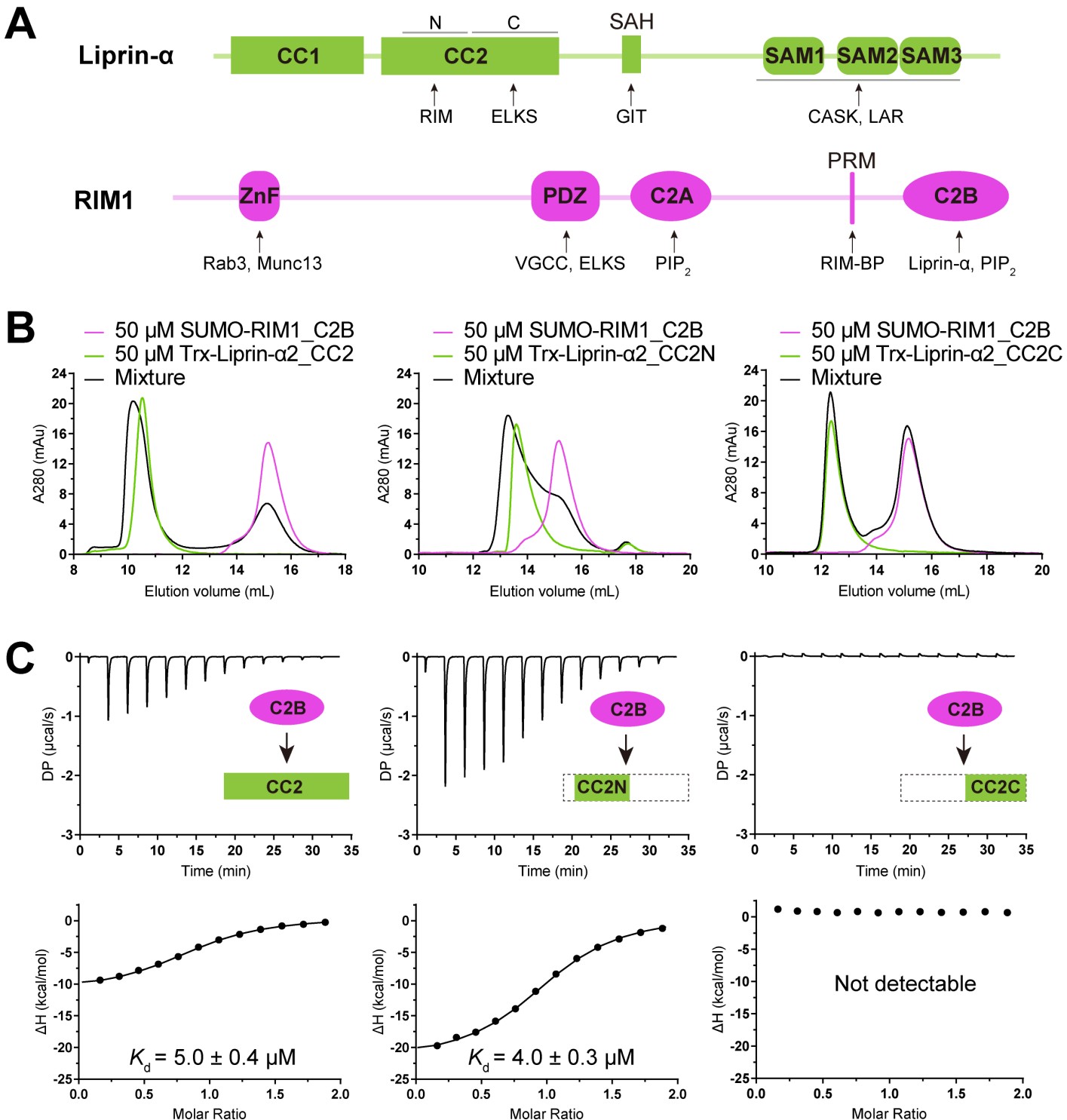

**Fig 1. Biochemical characterization of the liprin-α2/RIM1 interaction. (A)** Schematic of liprin-α and RIM1 domain organization, with protein-binding regions indicated. CC, Coiled-coil region; SAH, single alpha helix; SAM, sterile alpha motif; ZnF, zinc finger; PRM, proline-rich motif; PDZ, PSD-95/Discs-large/ZO-1 homology; C2, Protein kinase C conserved region 2. **(B)** aSEC analysis showing the binding of RIM1_C2B to the N-terminal segment of liprin-α2_CC2. **(C)** ITC-based affinity measurement of RIM1_C2B binding to different boundaries of liprin-α2_CC2. The data underlying panels B and C can be found in S1 Data.

In this study, we solved the structure of the complex formed by the coiled-coil region of liprin-α2 and the C2B domain of RIM1. The structure uncovers that liprin-α mutations associated with neurodevelopmental diseases block complex formation, supporting the importance of this complex for synapse physiology. Further structural analysis reveals a unique binding mode between the coiled-coil dimer and the C2B dimer, which drives a large protein assembly. Through this multivalent binding, liprin-α2 promotes the condensate formation of RIM1, confirming its role as a scaffold for early recruitment and assembly of active zone proteins. Using human neurons lacking all liprin-α (liprin-α qKOs), we showed that the liprin-α/RIM interaction is dispensable for synapse formation while being required for normal vesicle recruitment and synaptic transmission. Importantly, our LLPS assays and synaptic analyses indicate that liprin-α2, through its binding to RIM1, not only effectively accumulates RIM1 at the active zone but also promotes VGCC clustering, allowing proximal coupling between VGCC clustering sites and vesicle priming sites for efficient neurotransmitter release. Collectively, our study unveils the presynaptic assembly and regulatory mechanism of active zone machinery via the liprin-α/RIM interaction in an LLPS-dependent manner.

## Results

### Biochemical and crystallographic analyses of the liprin-α2/RIM1 complex

To understand the molecular mechanism governing the liprin-α/RIM interaction, we first identified the minimal segment in liprin-α2 that is sufficient for its binding to the C2B domain of RIM1. As our previous study suggests that liprin-α can form a tripartite complex with RIM and ELKS and a coiled-coil region (CC2; Fig 1A) binds both RIM and ELKS [23], we speculated that RIM and ELKS interact with distinct segments of CC2. To validate this, we divided the CC2 region into two parts: the N-terminal half (CC2N) and the C-terminal half (CC2C) (Fig 1A). The interactions between the two segments and RIM1_C2B were characterized using analytical size-exclusion chromatography (aSEC) and isothermal titration calorimetry (ITC). The results showed that CC2N but not CC2C interacts with RIM1_C2B, with a measured binding affinity of ~5 μM (Fig 1B and 1C). Consistently, the deletion of CC2N from liprin-α2 did not compromise its ability to bind ELKS1, while the deletion of CC2C disrupted the interaction (S1 Fig). These results indicate that liprin-α2 interacts with the C2B domain of RIM1 through its CC2N segment. The differential binding specificity found in the CC2 region provides an assembly mechanism for the liprin-α/RIM/ELKS tripartite complex.

Next, we aimed to solve the structure of the liprin-α2_CC2N/RIM1_C2B complex using X-ray crystallography. Initial attempts to co-crystallize the tag-removed CC2N and C2B fragments resulted in heavy precipitation, preventing crystal formation. To circumvent this issue, we modified our approach by mixing tag-removed liprin-α2_CC2N with SUMO-tagged RIM1_C2B for crystallization. However, this approach also failed to yield any crystals, despite extensive trials, presumably due to the interference of the SUMO tag in protein crystallization. To remove the SUMO tag without inducing severe protein precipitation, we added a trace amount of TEV protease during crystallization. This strategy successfully led to the formation of high-quality crystals, which we used to determine the structure of the liprin-α2_CC2N/RIM1_C2B complex at a resolution of 2.75-Å (Table A in S1 Text).

### Overall structure of the liprin-α2_CC2N/RIM1_C2B complex

In the complex structure, the CC2N segment forms a dimeric coiled coil that interacts with two RIM1_C2B molecules symmetrically through its N-terminal region, assembling a 2:2 stoichiometric complex (Figs 2A, S2A, and S2B). Conversely, the C2B domain of RIM1, characterized by a β-sandwich fold, mainly packs with CC2N via a β-sheet composed of strands β-2/3/6/9 (Figs 2A, 2B, and S2C). The CC2N/C2B interaction is predominantly mediated by polar interactions. Several salt bridges, including R1239[RIM1]-E328/D335[liprin-α2], R1201[RIM1]-E337[liprin-α2], and E1198[RIM1]-R346[liprin-α2], strongly stabilize this interaction (Fig 2B). Q332[liprin-α2] and R339[liprin-α2] also contribute significantly to the binding by forming hydrogen bond networks at the interface (Fig 2B). In addition to these polar interactions, hydrophobic interactions further strengthen the CC2N/C2B interaction (Fig 2B). The interface residues in CC2N are highly conserved across liprin-α isoforms in different species

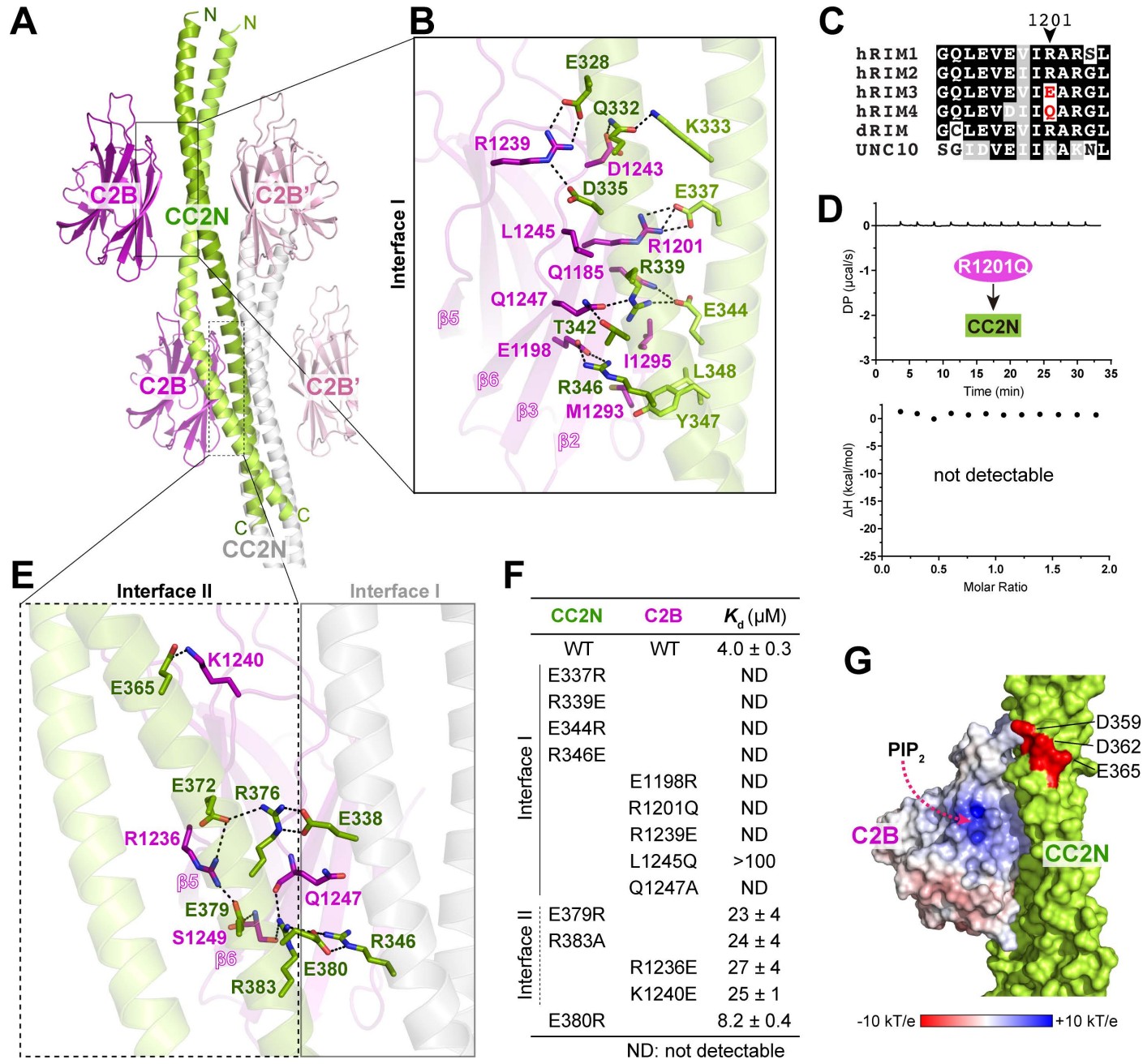

**Fig 2. Structural characterization of the liprin-α2_CC2N/RIM1_C2B complex. (A)** Crystal structure of the liprin-α2_CC2N/RIM1_C2B complex. Two neighboring CC2N coiled coils (colored green and gray, respectively), with four bound C2B molecules (colored magenta), are shown. **(B)** Molecular details of interface I formed between the N-terminal part of CC2N and C2B. Salt bridges and hydrogen bonds are indicated by dashed lines. **(C)** Multi-sequence alignment of RIM isoforms from various species, showing the sequence variability at the interface residue position corresponding to R1201 in hRIM1. Species abbreviations: 'h' for human, 'd' for *Drosophila*, and UNC10 as the *Caenorhabditis elegans* RIM homolog. **(D)** ITC analysis of the R1201Q RIM1 mutant's binding to CC2N. **(E)** Molecular details of interface II formed between the C-terminal part of CC2N and C2B. The interconnectivity between interfaces I and II in the complex of one C2B molecule and two CC2N coiled coils is displayed. Salt bridges and hydrogen bonds are indicated by dashed lines. **(F)** Summary of binding affinities between various CC2N and C2B variants, measured by ITC. **(G)** Surface representation of C2B and CC2N, showing the spatial relationship between the $PIP_2$-binding site and bound CC2N. Key negatively charged residues on the CC2N structure are highlighted in red. The data underlying panel D can be found in S1 Data.

(S2B and S2D Fig), suggesting that the observed RIM-binding mode is shared by all liprin-α proteins. Conversely, a key interface residue in RIM1, R1201, is not conserved in RIM3 and RIM4 (Fig 2C). The substitution of R1201 with glutamine in RIM1, to mimic the sequence of RIM4, abolished the CC2N/C2B interaction (Fig 2D), confirming that the liprin-α/RIM interaction is specific to certain RIM proteins [29], including RIM1, RIM2, and their homologs in invertebrates. Likewise, the presence of a glutamine residue at the R1201-corresponding position in the C2A domain of RIM1 explains the selective binding of liprin-α to the C2B domain over C2A (S2C Fig).

In addition to the primary interface (interface I) in the formation of the tetrameric complex, the tetramers in the crystal are assembled through a secondary CC2N/C2B interface (interface II) (Fig 2A). At interface II, the C-terminal part of CC2N interacts with a side face of the β-sandwich fold in RIM1_C2B, mainly through charge-charge interactions and hydrogen bonding (Figs 2E, S2B, and S3A). ITC-based analysis showed that, while disruptive mutations at interface I abolish the CC2N/C2B interaction, interface II mutations have a milder impact, reducing the binding affinity by ~5-fold (Figs 2F, S3B, and S3C). Furthermore, neighboring CC2N coiled coils interact through salt bridges that stabilize both CC2N/C2B interfaces (Figs 2E and S3A). Specifically, E380 forms salt bridges with R346 in a neighboring CC2N coiled coil, stabilizing the orientation of R346 for its binding to E1198$^{RIM1}$ at interface I (S3D Fig). The charge-reversed mutation E380R led to a 2-fold decrease in the binding affinity between CC2N and C2B (Figs 2F and S3E). Together, the crystal structure of the liprin-α2_CC2N/RIM1_C2B complex reveals how the CC2N coiled coil specifically recognizes the C2B domain of RIM1 via two interconnected binding interfaces.

As the C2B domain of RIM1 also binds to phosphatidylinositol 4,5-bisphosphate (PIP$_2$) [29], we analyzed the potential impact of PIP$_2$ on liprin-α's ability to bind C2B. As shown in Fig 2G, the putative PIP$_2$-binding site on C2B remains fully accessible with bound CC2N. However, when C2B is associated with the PIP$_2$-containing membrane, formation of the liprin-α/RIM complex positions a negatively charged patch on the CC2N surface facing the membrane (Fig 2G). Considering the negatively charged nature of the inner leaflet of the plasma membrane, this spatial arrangement has the potential to generate charge repulsion, thus inhibiting the CC2N/C2B interaction. It suggests that the membrane association of RIM1 in the PIP$_2$-enriched compartment may tune its binding to liprin-α.

## Two disease-associated mutations at the CC2N region disrupt the liprin-α/RIM interaction

Many genetic mutations in human liprin-α genes have been linked to neurodevelopmental disorders such as autism, intellectual disability, and epilepsy [33–36]. Interestingly, several reported missense mutations are located in the CC2N region (Fig 3A) [33,35–37]. These mutated sites, including E328$^{liprin-α2}$, A315$^{liprin-α3}$ (corresponding to A350 in liprin-α2), and L330$^{liprin-α1}$ (L348 in liprin-α2), are strictly conserved in the liprin-α family (Fig 3B). In addition to E328's critical role in forming salt bridges with RIM1_C2B (Fig 2B), L348 is directly involved in hydrophobic interactions with M1293 and I1295 in RIM1_C2B (Fig 2B). A350, although not directly involved in C2B binding, contributes to the coiled-coil formation of CC2 (Figs 3C and S2B). The charge-reverse mutation E328K disrupts the charge-charge interaction between liprin-α2 and RIM1. To determine the potential consequence of the other two mutations, we performed *in silico* substitutions of the corresponding residues in the complex structure and analyzed the mutated model. As shown in Fig 3D, the L348F mutation, while retaining hydrophobicity, imposes steric hindrance by introducing a bulkier sidechain, thereby impeding the close contact between CC2N and C2B. However, the A350S mutation, having little impact on the coiled-coil structure of CC2N (Fig 3E), is unlikely to interfere with the CC2N/C2B interaction.

To further evaluate the mutational effects on the liprin-α/RIM interaction, we introduced E328K, L348F, and A350S mutations to the CC2N construct and measured the binding affinities of the CC2N mutants to RIM1_C2B. Consistent with our structural analysis, the E328K and L348F mutations eliminate the CC2N/C2B interaction, while the A350S mutation had minimal impact on binding affinity (Fig 3F). Considering that the L330F mutation in liprin-α1 was identified in patients with autism [35], our results suggest that this mutation may impair synapse development by interfering with the binding of liprin-α1 to RIM proteins. Nevertheless, given the poorly defined function of liprin-α1 at the presynapse, whether the L330F

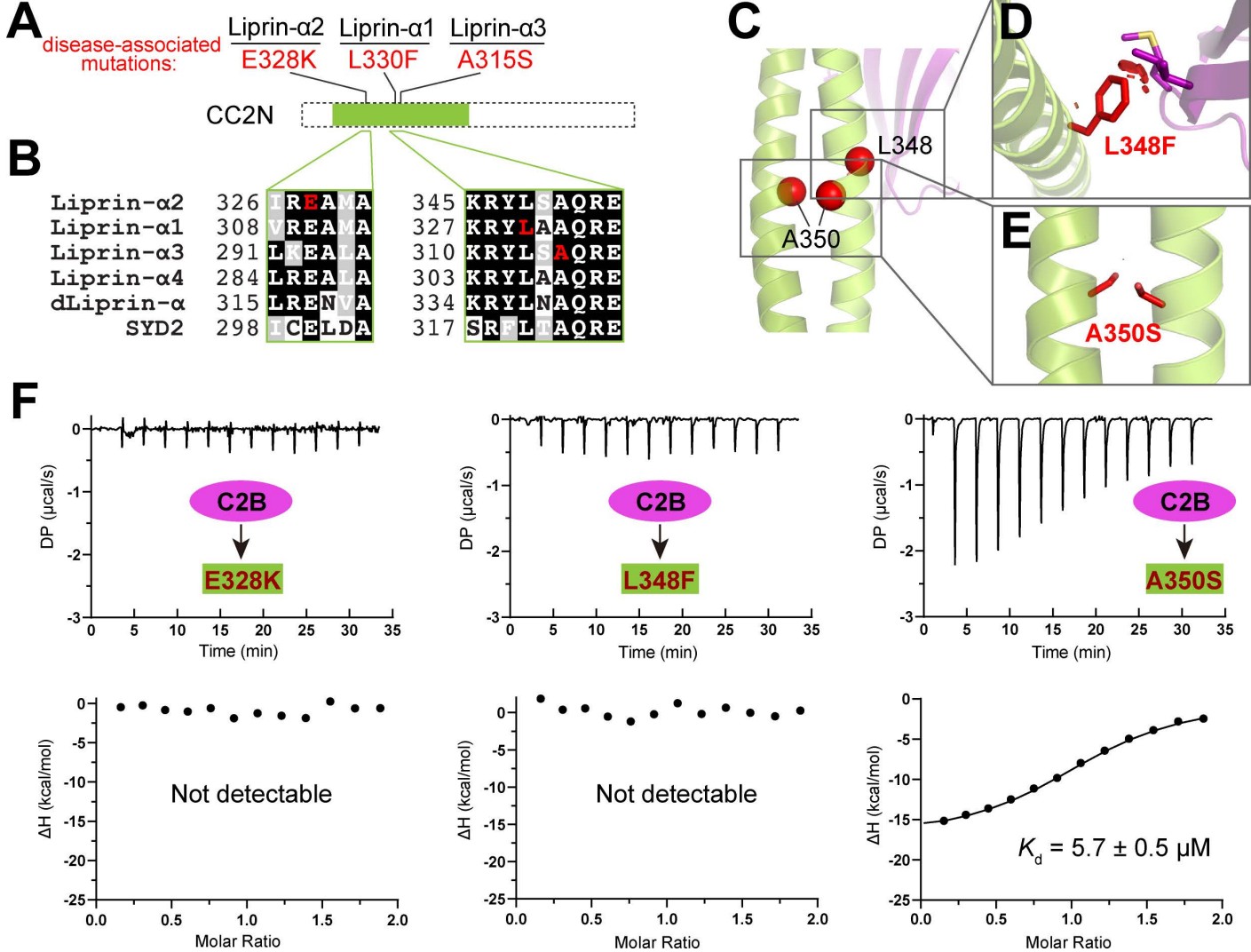

**Fig 3. Structural and biochemical analyses of disease-associated mutations on the liprin-α/RIM interaction. (A)** Disease-associated mutations and their positions in the CC2N segment of liprin-α1 (L330, corresponding to L348 in liprin-α2), liprin-α2 (E328), and liprin-α3 (A315, corresponding to A350 in liprin-α2). **(B)** Multisequence alignment of liprin-α family members. Residues affected by disease-associated missense variants are marked in red. **(C)** Cartoon representation of the liprin-α2_CC2N/RIM1_C2B complex with residues affected by disease-associated missense variants indicated. E328 and L348 are located at interface I, while A350 contributes to the coiled-coil formation of CC2N. **(D)** Structural analysis of the L348F mutation showing steric hindrance caused by the mutated sidechain upon the complex formation. Atomic clashes are indicated by red cylinders. **(E)** Structural analysis of the A350S mutation in the context of the coiled-coil structure, revealing no disruption. **(F)** ITC-based analyses of the interactions between C2B and the E328K, L348F, and A350S mutants of CC2N. The data underlying panel F can be found in S1 Data.

mutation contributes to neurodevelopmental defects through the disruption of liprin-α1/RIM interaction requires further investigation.

## Liprin-α2 and RIM1 can form a large complex through multivalent binding in vitro

Although RIM1_C2B was purified as a monomer, it forms a homodimer in our crystal structure, a feature also observed in the apo C2B structure of RIM1 [38] (Figs 4A and S4A). This dimerization tendency was confirmed in solution, as increasing concentrations of RIM1_C2B led to a corresponding increase in dimer formation (Figs 4B and S4B). Considering the

C2B dimerization along with the CC2N/C2B interactions, we propose that liprin-α2 and RIM1 may assemble into a large complex through a network of intermolecular interactions revealed in our crystal structure (Figs 4C and S4A). This hypothesis is supported by aSEC analysis of a 500 μM liprin-α2_CC2/RIM1_C2B mixture, which showed the formation of protein assemblies with molecular weights even larger than 500 kDa (Fig 4D). In contrast, either CC2 or C2B alone maintains their dimeric state even at a concentration of 500 μM (S4B and S4C Fig). These results suggest the involvement of multivalent interactions in assembling the liprin-α2/RIM1 complex. However, given the high protein concentrations (sub-mM) used in our crystallographic and aSEC analyses, it prompts an important question of whether the observed multivalent interactions could occur in the context of presynaptic assembly.

Recent studies on active zone proteins, including liprin-α, RIM, and ELKS, reveal their pronounced propensity for LLPS [23,30,32,39]. Through LLPS, these proteins can concentrate within condensates ranging from sub-mM to mM levels [23,30]. Given the well-established significance of multivalent binding in mediating LLPS [40,41], it is plausible that liprin-α2 and RIM1 may coalesce into a co-condensate, presumably facilitated by the multivalent interactions identified in our crystal structure. To explore the potential role of the CC2N/C2B interaction in the co-condensation of liprin-α2 and RIM1, we performed in vitro LLPS assays with a purified N-terminal segment of liprin-α2 (liprin-α2_CC12) and full-length RIM1. Notably, liprin-α2_CC12, containing both the CC1 and CC2 regions, has been shown to promote the LLPS of ELKS [23]. Indeed, compared to the condensate formed by RIM1 alone, the addition of liprin-α2_CC12 robustly enlarged the RIM1 droplet size (Figs 4E, 4F, and S4D). In contrast, the E334R and R346E mutations at interface I of CC12 diminished the promotive effect, confirming the importance of the CC2N/C2B interaction in promoting RIM1 LLPS. Additionally, the R383A mutation at interface II of CC12 modestly attenuated the enhancing effect on droplet size (Fig 4E and 4F), consistent with the milder impact of interface II on the disruption of the CC2N/C2B interaction (Fig 2F). Similarly, our in vitro sedimentation-based assay confirmed the CC12-mediated promotion effect of RIM1 LLPS (S4E and S4F Fig). Altogether, our structural and biochemical analyses highlight the role of the CC2N/C2B interaction in assembling liprin-α2 and RIM1 into condensates, offering insights into the molecular mechanism underlying the dynamic assembly of the presynaptic active zone.

## The CC2N/C2B interaction is dispensable for synaptic formation

To study the functional role of the liprin-α2/RIM1 complex in active zone assembly and function, we employed pluripotent stem-cell-derived human neurons lacking all liprin-α proteins (liprin-α qKO neurons) [11], in which the assembly of active zones and the recruitment of synaptic vesicles to nascent terminals is completely blocked. By re-expressing wild-type (WT) liprin-α2 or its mutants, including the CC2N deletion (ΔCC2N) and point mutations E344R and R346E, we can specifically analyze the CC2N/C2B interaction's effects on synaptic defects in liprin-α qKO neurons, as these mutants disrupt RIM1 binding while retaining ELKS1 interaction (S5A and S5B Fig). To ensure comparable expression levels of liprin-α2 variants, an approach of lentivirus transduction was used (S5C Fig).

Quantitative analysis of synapses using synapsin and MAP staining for pre- and post-synaptic compartments, respectively, showed that the deletion of all liprin-α isoforms almost eliminated synaptic puncta as we reported previously [11], but re-expression of either liprin-α2 WT or RIM-binding-deficient mutants comparably restored synapsin puncta signals in liprin-α qKO neurons (S6A and S6B Fig). This finding suggests that the disruption of the liprin-α2/RIM1 complex has no major effects on active zone assembly, and the CC2N/C2B interaction is thus dispensable for synapse formation.

## The CC2N/C2B interaction promotes neurotransmitter release

To further assess the significance of the liprin-α/RIM complex on synaptic functions, we performed measurements of miniature excitatory postsynaptic currents (mEPSCs) in liprin-α qKO neurons. The absence of all four liprin-α isoforms nearly abolished spontaneous synaptic transmission (Fig 5A and 5B), consistent with our previous findings [11]. We then rescued the synaptic deficit in liprin-α qKO neurons by re-expressing liprin-α2 or its mutants. As shown in Fig 5A and 5B,

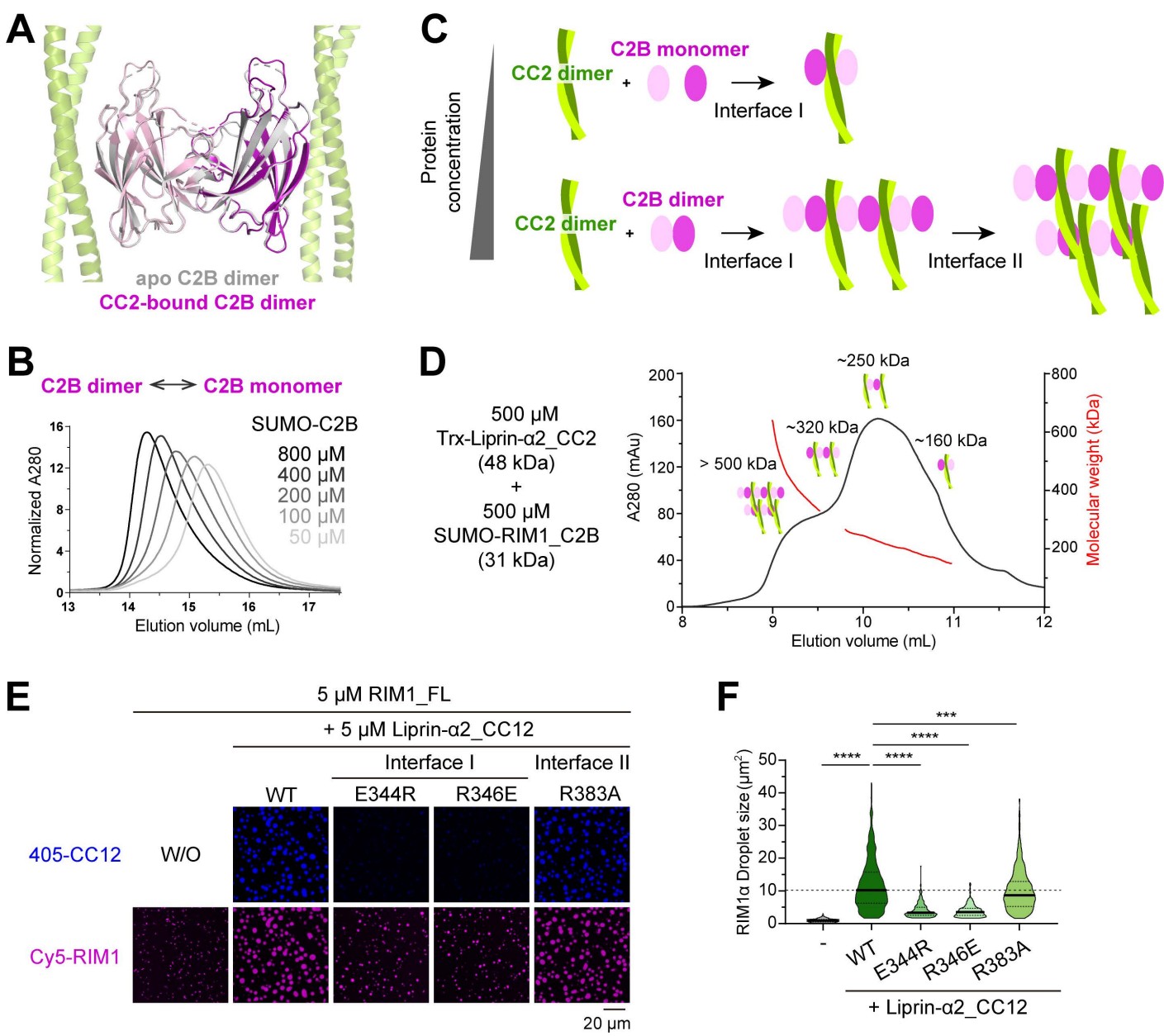

**Fig 4. Unique assembly mode of the liprin-α2/RIM1 complex. (A)** Structural superimposition of the RIM1_C2B dimer from two crystal structures. The overall RMSD of these two dimeric C2B structures is 0.4 Å. **(B)** Concentration-dependent dimerization of RIM1_C2B in solution. **(C)** Schematic representation of the liprin-α2_CC2N/RIM1_C2B complex assembly modes. Under low-concentration conditions, a 2:2 heterodimer is formed, whereas high-concentration conditions lead to the assembly of a large complex through multiple intermolecular interactions. **(D)** aSEC analysis coupled with multi-angle static light scattering (MALS), showing the formation of large CC2N and C2B assemblies. **(E)** In vitro LLPS assays showing the CC2N/C2B assembly in promoting RIM1 LLPS. Confocal images were captured 20 min after the mixing of RIM1 and CC12 or its mutants. The time course of condensate formation following the mixing is also shown in S4D Fig. **(F)** Quantification analysis of RIM1 droplet sizes presented in panel **E**. Droplets from eight different views are quantified, and all data are represented as means±SD. The unpaired Student $t$ test analysis was used to define a statistically significant difference (****$p < 0.0001$; ***$p < 0.001$). The data underlying panels B and D can be found in S1 Data.

WT liprin-α2 re-expression substantially restored mEPSC frequency, whereas the mutants were significantly less effective. However, the amplitude or kinetic properties of mEPSCs were not affected by the re-expression of mutant liprin-α2 (Fig 5C), indicating that neither the amount of transmitter loaded in synaptic vesicles nor the fusion kinetics of spontaneous release events were affected upon disruption of liprin-α/RIM interactions. These results indicate that liprin-α/RIM complexes promote neurotransmission by regulating the extent but not the speed of synaptic vesicle fusion from presynaptic terminals.

Next, we explored whether the liprin-α/RIM complex influences neurotransmission by regulating the number of primed synaptic vesicles. For this, we analyzed the size of the readily releasable pool (RRP) of synaptic vesicles, using hyperosmotic sucrose treatment as described earlier [42]. In liprin-α qKO neurons, sucrose responses were eliminated but were readily rescued by liprin-α2 WT re-expression (Fig 5D–5F). In contrast, RIM-binding-deficient mutants only partially rescued sucrose responses, which were significantly smaller than those triggered by the WT rescue (Fig 5D–5F). These observations suggest that the liprin-α/RIM interaction is important for maintaining the RRP size. Disruption of liprin-α/RIM complexes may thus reduce the number of primed vesicles, which could, at least in part, explain the observed reduction in mEPSC frequency described above (Fig 5A and 5B). Together, our structural and functional results suggest that the liprin-α/RIM complex, assembled by the CC2N/C2B interaction, controls the synaptic transmission, at least in part, by regulating the number of primed synaptic vesicles in nerve terminals.

### Liprin-α2 facilitates the presynaptic accumulation of RIM1 through the CC2N/C2B interaction

Given the essential role of RIM1 in vesicle priming at the active zone [43,44], the liprin-α/RIM1 complex may regulate RIM1 levels at the active zone, which in turn could influence synaptic vesicle priming and release. To test this possibility, we analyzed the presynaptic level of RIM1 using STED super-resolution microscopy (S6C Fig). For this, we first searched for "synapse-rich" regions as identified by the high density of Synapsin/PSD95 appositions, which represent fully assembled pre-post junctions. We then measured RIM1 signals in these regions and found a significant reduction of RIM1 signals under the R346E rescue condition, compared to the WT liprin-α2 (Fig 5G). This finding, coupled with the role of liprin-α2 in promoting RIM1 condensate formation through the CC2N/C2B interaction (Fig 4E and 4F), suggests that liprin-α may effectively accumulate RIM1 at the active zone by forming the liprin-α/RIM1 complex. As RIM1 may also be involved in recruiting VGCCs to the active zone [26,45], we used STED microscopy to measure the active zone levels of P/Q-type VGCC α1 subunit (CaV2.1) and compared them between the WT and R346E rescue conditions. However, no significant alternation in CaV2.1 levels was detected (Fig 5H), suggesting that while the liprin-α/RIM interaction helps recruiting RIM1 to nerve terminals, it does not control the overall density of VGCCs at the active zone. Nevertheless, potential nanoscale changes in $Ca^{2+}$ channel positioning within the active zone upon liprin-α/RIM interaction disruption cannot be ruled out. Indeed, a previous study of the calyx of Held synapses has clearly shown changes in presynaptic VGCC nano-organization without observing changes in the overall density of these channels at the nerve terminals [46].

### The liprin-α/RIM complex controls the clustering of $Ca^{2+}$ channels through mesoscale interactions among ELKS1 and RIM1 condensates

Liprin-α, through its CC2 region, assembles ELKS and RIM proteins (Fig 6A), which contribute to the nano-scale organization of presynaptic VGCCs [26,47]. Therefore, we hypothesize that the liprin-α/RIM complex may cooperate with ELKS to regulate the local distribution of VGCCs, without affecting the overall level of $Ca^{2+}$ channels at the active zone. Intriguingly, both ELKS1 and RIM1 can form condensates via LLPS, regulated by liprin-α and RBP, respectively [23,30]. To explore the relationship between the ELKS1 and RIM1 condensates, we prepared these condensates using purified full-length proteins. To enhance RIM1 LLPS, the $RBP2\_(SH3)_3$ fragment was added (Fig 6A), as reported previously [30]. Without liprin-α2_CC12, the ELKS1 and RIM1 condensates merge to form co-phase droplets (Fig 6B). However, the presence of liprin-α2_CC12 prevents co-condensation, with the droplets of ELKS1 and RIM1 become largely immiscible (Fig 6C).

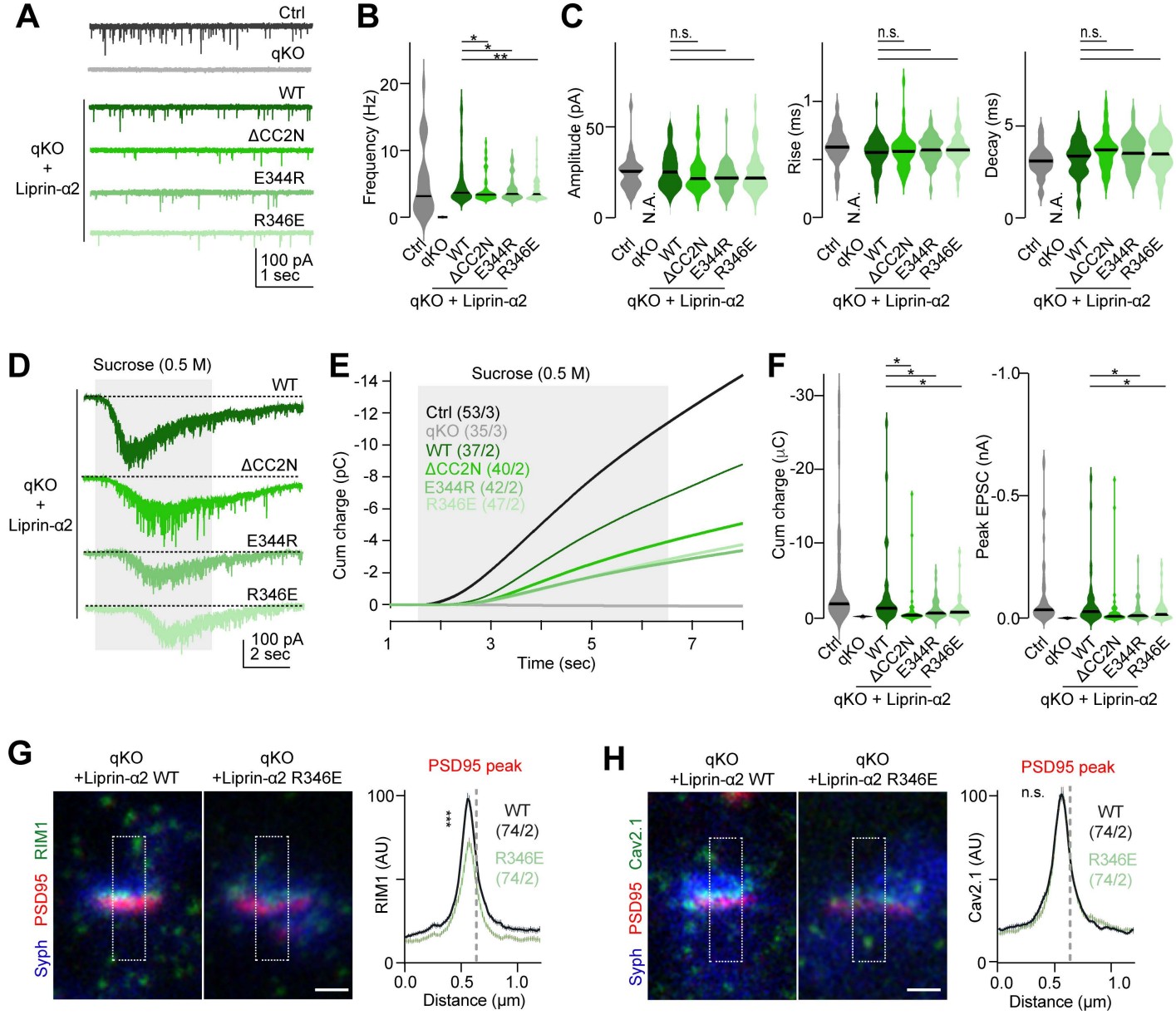

**Fig 5. Liprin-α2/RIM1 complex controls synaptic function. (A)** Representative traces of spontaneous miniature excitatory postsynaptic currents (mEPSCs) in Control (Ctrl) neurons, liprin-α quadruple knockout (qKO), and qKO neurons rescued with liprin-α2 WT or RIM-binding-deficient mutants (ΔCC2N, E344R, and R346E). Recordings were performed at a holding potential of −70 mV and in the presence of 0.5 µM tetrodotoxin (TTX). **(B)** Summary plots of mEPSC frequency under the conditions indicated in panel **A**. Data are represented as means±SD. Number of experiments analyzed (cells/batches): Ctrl: 30/3, Ctrl: 30/3, qKO: 35/3, liprin-α2 WT: 78/3, liprin-α2 ΔCC2N: 54/3, liprin-α2 E344R: 54/3, liprin-α2 E346E: 60/3. *$p < 0.05$. **(C)** Summary plots of mEPSC properties under the conditions indicated in panel **A**. From left to right, the amplitude, rise time, and decay time of mEPSCs are shown. Data are represented as means±SD. n.s. non-significant. **(D)** Representative traces showing the response to hyperosmotic sucrose in indicated conditions. Neurons were challenged with 0.5 M sucrose for 5 s (shaded area in panel **E**) using a flow pipe placed in close proximity (near 100 µm) to the recorded cells. **(E)** Integrated responses (EPSC charge) to hypertonic sucrose application in indicated conditions. **(F)** Statistical analyses of the EPSC charge (left) and peak amplitude (right) of sucrose-evoked responses, showing the impact of liprin-α2 mutations on the size of the RRP. Data are represented as means±SD. The number of cells/batches analyzed for each condition was indicated in panel **E**. *$p < 0.05$. **(G, H)** Subsynaptic imaging and summary plots of RIM1 **(G)** and Cav2.1 **(H)** intensity profiles in liprin-α qKO neurons rescued with liprin-α2 WT (black) or R346E mutant (light green). The relative peak of the PSD95 signal is indicated by the vertical dotted line. Scale bar: 250 nm. Number of profiles/batches analyzed: liprin-α2 WT:74/2; liprin-α2 R346E:74/2. Data represented as means±SEM; n.s. non-significant, ***$p < 0.001$. The data underlying this figure can be found in S1 Data.

Consistent with our previous findings using ELKS2 fragments [23], ELKS1/liprin-α2_CC12 was found to enrich at the periphery of RIM1 droplets (Fig 6C, box 'a'). Conversely, no RIM1 accumulation was observed in ELKS1 droplets (Fig 6C, box 'b'), although both liprin-α2_CC12 and ELKS1 can interact with RIM1 (Fig 6A).

The disruption of the liprin-α/RIM interaction by introducing R346E into liprin-α2_CC12 eliminated the surface coating of CC12 on RIM1 droplets (Fig 6C), indicating that the liprin-α/RIM interaction is essential for recruiting CC12 onto RIM1 condensates. However, the absence of liprin-α2_CC12 from RIM droplets led to an increased accumulation of ELKS1 within RIM1 droplets, in line with the formation of ELKS1/RIM1 co-phase droplets in the absence of CC12 (Fig 6B). These observations suggest that the liprin-α/RIM interaction limits the accumulation of ELKS1 within the RIM1 condensate. Additionally, without the addition of RBP2_(SH3)$_3$, RIM1, ELKS1, and liprin-α2_CC12 form co-phase droplets (S7A Fig), indicating that RBP2 also contributes to the immiscibility between the RIM1 and ELKS1 condensates when liprin-α2_CC12 is present. Consistently, liprin-α2_CC12 is weakly enriched on the periphery of RIM1/RBP2_(SH3)$_3$ co-condensates (S7B Fig), compared to its co-condensation formation with RIM1 alone (Fig 4E). Thus, the peripheral enrichment of liprin-α2_CC12 changes the surface property of the RIM1 condensates, which potentially hinders the diffusion of ELKS1 molecules into condensates. This hypothesis is supported by fluorescence recovery after photobleaching (FRAP) analyses, which show a decrease in the dynamic properties of RIM1 condensates in the presence of liprin-α2_CC12 compared to RBP2_(SH3)$_3$ (S7C Fig).

The RIM1 condensate is known to enrich the cytoplasmic tail of presynaptic Ca$^{2+}$ channels [23,30], which interacts with the PDZ domain of RIM (Fig 6A) [26]. As ELKS interacts with the PDZ domain of RIM with a much higher binding affinity [24,25,30,48], the accumulation of ELKS1 in RIM1 condensates may prevent the enrichment of VGCCs through binding competition. Indeed, the addition of the cytoplasmic tail of the N-type VGCC α1 subunit (NCav_CT) to the mixture of ELKS1 and RIM1 condensates resulted in its accumulation in RIM1 condensates (Fig 6D). However, disrupting the liprin-α/RIM interaction decreased the overall level of NCav_CT in RIM1 condensates, presumably due to the accumulation of ELKS1 in RIM1 condensates (Fig 6D). By classifying droplets based on their fluorescence intensity relationship (Fig 6E), we quantitatively compared the ELKS1 and NCav_CT intensities in the two classes of condensates under different conditions (Figs 6F and S7D). The results confirmed that disrupting the liprin-α/RIM interaction significantly increases the distribution of ELKS1 in RIM1 condensates (Fig 6F, left panel), which in turn dramatically reduces the distribution of NCav_CT in RIM1 condensates to a level comparable to that in ELKS1 condensates (Fig 6F, right panel).

To further validate our findings, we purified full-length liprin-α2 with reasonable quality (S8A Fig). Consistent with our previous observations using protein fragments [23], the addition of full-length liprin-α2 to ELKS condensates dramatically expanded the droplet size (S8B Fig), indicating that ELKS forms co-condensates with liprin-α2. We then added these ELKS/liprin-α2 co-condensates into pre-formed RIM1/RBP2 condensates. As expected, full-length liprin-α2 effectively restricted the entry of ELKS into RIM/RBP condensates (S8C Fig), comparable to the impact observed with the CC12 fragment (Fig 6C), further validating our findings based on the CC12 fragment. Taken together, our LLPS analyses demonstrate that the liprin-α/RIM complex plays a regulatory role in the distribution of both ELKS1 and VGCCs in the RIM1 condensate. Considering the essential role of VGCC nano-scale clustering in efficient vesicle release, liprin-α may control the interplay between RIM, ELKS, and VGCCs in synaptic transmission by regulating mesoscale protein–protein interactions in the condensed phase.

## The liprin-α/RIM complex promotes efficient coupling between presynaptic Ca$^{2+}$ channels and primed synaptic vesicles

We next directly assessed if liprin-α/RIM complex, by regulating the accumulation of ELKS1 in RIM1 condensates, can control the fine-scale localization of VGCCs within the active zone and thereby regulate the nanodomain coupling between VGCCs and primed synaptic vesicles. For this, we used a channel-rhodopsin-assisted approach to assess evoked synaptic transmission in liprin-α qKO neurons rescued with either liprin-α2 WT or R346E (Fig 7A). This approach

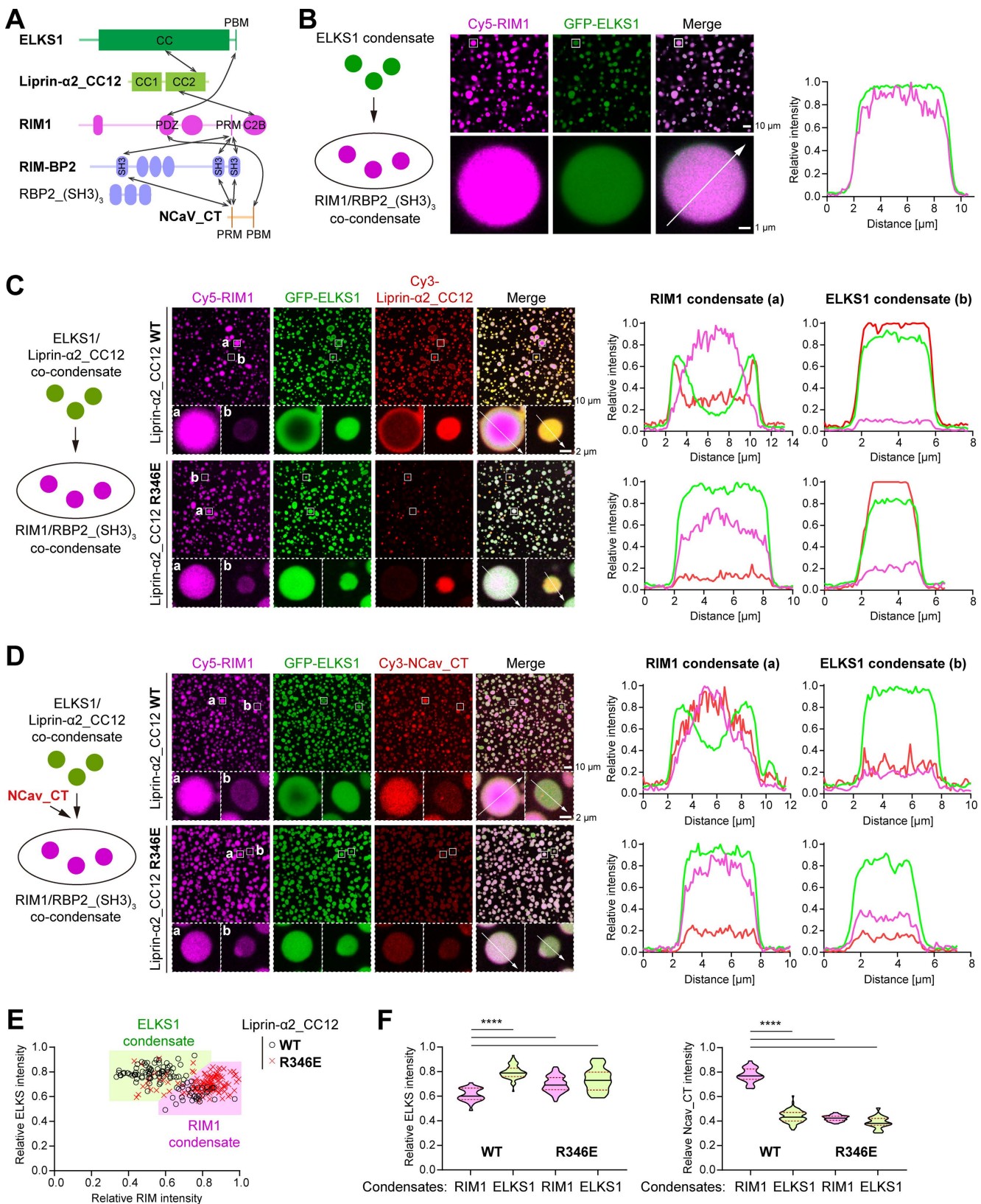

**Fig 6. The liprin-α/RIM interaction modulates the distribution of ELKS1 and NCav_CT in RIM1 condensates. (A)** Schematic diagram of the interaction network among active zone proteins. Interactions are indicated by the double-headed arrows. **(B)** Confocal imaging analysis of the LLPS mixture containing the ELKS1 condensate and the RIM1/RBP2_(SH3)$_3$ condensate. A magnified view of a representative droplet was displayed below, with a line analysis of fluorescence signal intensities along the indicated line. The concentration of each protein was 5 μM. **(C)** Confocal imaging analysis of the LLPS mixture containing the ELKS1/liprin-α2_CC12 (WT or R346E) condensate and the RIM1/RBP2_(SH3)$_3$ condensate. Magnified views of representative RIM1 (a) and ELKS1 (b) droplets were displayed below, with line analyses of fluorescence signal intensities along the indicated lines. The concentration of each protein was 5 μM. **(D)** Confocal imaging analysis of the LLPS mixture containing the ELKS1/liprin-α2_CC12 (WT or R346E) condensate, the RIM1/RBP2_(SH3)$_3$ condensate, and NCav_CT. Magnified views of representative RIM1 (a) and ELKS1 (b) droplets were displayed below, with line analyses of fluorescence signal intensities along the indicated lines. The concentration of each protein was 5 μM. **(E)** Plot analyses of the intensity relationship between ELKS1 and RIM1 fluorescence signals in the condensates shown in panel **D.** The intensity of ~100 droplets in the view was quantified and normalized. The relative intensity ratio of RIM1/ELKS1 >1 was defined as RIM1 condensate, while the ratio <1 was defined as ELKS1 condensate. **(F)** Quantitative analyses of RIM1, ELKS1, and NCav_CT fluorescence intensities in the ELKS1 and RIM1 condensates. Data represented as means±SD. The unpaired Student *t* test analysis was used to define a statistically significant difference (****$p$<0.0001). The data underlying panels B-F can be found in S1 Data.

represents a straightforward method to record the amplitudes of evoked synaptic transmission in cultured neurons [11,46]. Nevertheless, it precludes accurate measurements of the kinetics of the evoked release, due to the limited efficacy of light in triggering spikes with as high temporal accuracy as for extracellular electrical stimulation. We loaded nerve terminals with the 'slow' $Ca^{2+}$-chelator EGTA-AM, which selectively chelates diffusing $Ca^{2+}$ ions not involved in nanodomain coupling between $Ca^{2+}$ channels and the release machinery (Fig 7B). We reasoned that if the liprin-α/RIM complex is critical for this short-distance coupling, evoked release in the R346E rescue would be more sensitive to EGTA-AM than that in the WT rescue. Indeed, we observed that EGTA-AM was significantly more effective at blocking the amplitude of evoked release in liprin-α qKO neurons rescued with R346E, compared to those rescued with WT construct (Fig 7C and 7D), suggesting an increased distance between $Ca^{2+}$ channels and $Ca^{2+}$ sensors in the release machinery upon disruption of the liprin-α/RIM interaction. Control treatment with vehicle (DMSO) showed no significant differences in evoked release (Fig 7C and 7D). Altogether, these results indicate that the liprin-α/RIM complex is crucial for localizing $Ca^{2+}$ channels in close proximity to primed synaptic vesicles within the active zone and ensuring tight coupling between presynaptic action potentials and neurotransmitter release. These finding also provides a plausible mechanism for liprin-α/RIM complexes in the regulation of the nanoscale VGCC clustering, while having a negligible impact on its overall presynaptic levels (Fig 5H) that may involve alternative interactions between VGCC and other active zone components.

## Discussion

Our study offers insights into the assembly mechanism of the liprin-α/RIM complex, a critical yet less characterized interaction in the presynaptic active zone. Through a combination of structural biology and biochemistry, we uncover the sophisticated molecular interactions that govern the formation of this complex. Furthermore, our study elucidates how the liprin-α/RIM interaction regulates the interplay between the RIM1 and ELKS1 condensates in vitro, providing a plausible explanation for how active zones are organized at the nanometer scale. Last, using synaptic analyses in liprin-α qKO neurons with rescue assays, we studied the functional significance of this interaction and found that it regulates RIM distribution, vesicle priming, and VGCC clustering. Thus, our work deepens the understanding of the active zone's assembly mediated by liprin-α and RIM, as well as the dynamic coupling between protein machinery for vesicle priming and release in synaptic transmission.

Our results indicate that liprin-αs, via direct interactions with RIM, can dynamically control two essential functions of the active zone, namely synaptic vesicle priming and subsynaptic distribution of VGCCs, although via different mechanisms. The defects in vesicle priming observed upon disruption of liprin-α/RIM complexes (Fig 5) can be readily explained by a concomitant reduction in the levels of RIM1 at the active zone, known to regulate the size of the RRP in conjunction with Munc13. However, it is important to note that our measurements of vesicle priming are based on the use of hypertonic sucrose only, and would need to be directly confirmed in future studies using alternative approaches, such as

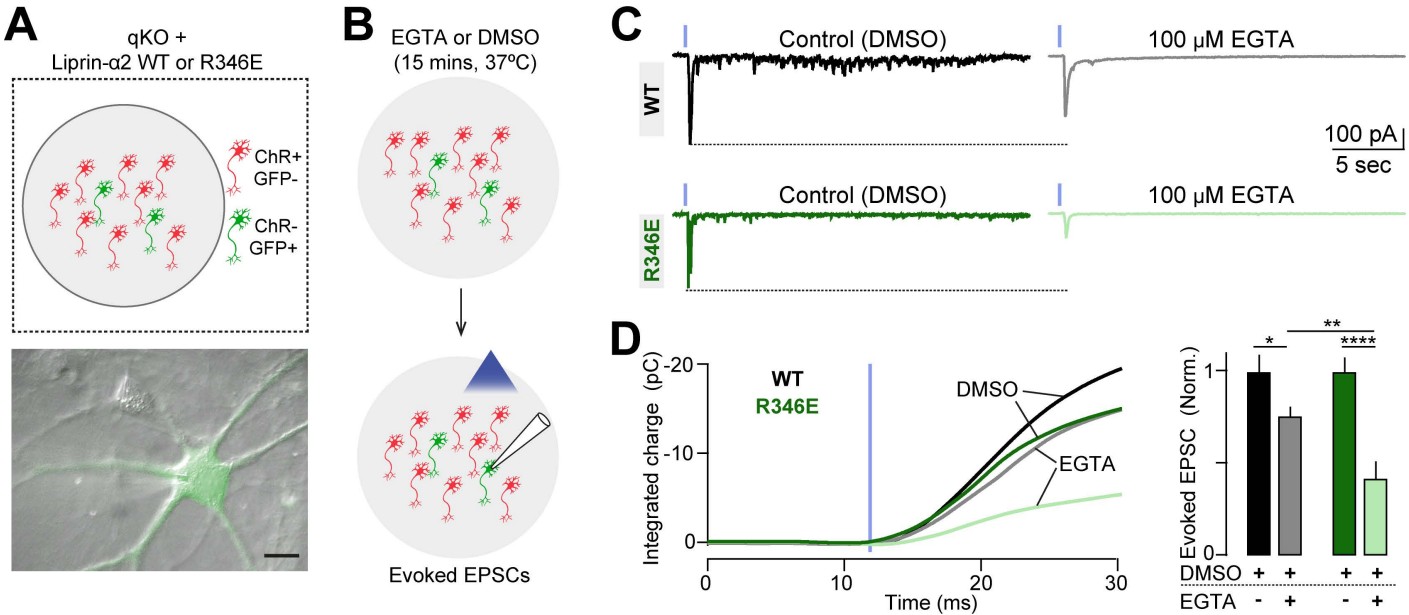

**Fig 7. Liprin-α/RIM complexes couple presynaptic Ca²⁺ channels with primed synaptic vesicles.** The data underlying panels C and D can be found in S1 Data.

high-pressure freezing electron microscopy. In contrast to vesicle priming, the defects in VGCC nanoscale localization (Figs 6 and 7) can be explained in part by the reduction of active zone RIM but perhaps more importantly by a redistribution of ELKS1 in RIM1 condensates, which can outcompete and displace VGCC bound to RIM PDZ domains, resulting in a dramatic reduction of VGCC levels in RIM1 condensates. In neurons, however, we failed to observe a significant reduction in the levels of VGCC levels at the active zone, using STED microscopy (Fig 5H). This suggests that in nerve terminals, the liprin-α/RIM complexes do not regulate the overall number of calcium channels at the active zone, but rather their fine-scale localization in close proximity to the synaptic vesicles, similar to the role played by RBPs at the calyx of Held synapses [46]. Future studies using higher resolution microscopy approaches, such as STORM, might help uncover the sub-active zone distribution of VGCC and the role of liprin-α/RIM complexes in defining it. Altogether, our results align with the idea that liprin-αs act as master organizers of presynaptic assembly[11,49], regulating the nano-organization of the active zone functions via a multitiered interaction network. At the core, liprin-αs can dynamically and directly interact with RIM and ELKS in a single protein complex, which further coordinates other presynaptic components such as Munc13, RBP, and calcium channels.

Our findings support the involvement of LLPS in the dynamic organization of presynaptic active zones. Emerging evidence has highlighted the nanoscale clustering of RIM [50], neurexin [51], Munc13 [52], and Ca²⁺ channels [53,54] at subregions of the active zones. Considering the active zone's capacity to adapt to various stimuli, LLPS likely facilitates the rapid reorganization of these presynaptic nanoclusters, enabling swift adjustments to synaptic activity. Mesoscale interactions between membrane-associated condensates (e.g., ELKS1 and RIM1 condensates) may contribute to such a dynamic organization of these presynaptic nanoclusters, providing mechanistic insights into the adaptable assembly of the active zone. In this framework, liprin-α emerges as a central regulatory hub, controlling the interplay between active zone proteins in the context of LLPS. Specifically, the liprin-α/RIM interaction not only promotes the recruitment of RIM1 to the active zone (Fig 5G) but also restricts the enrichment of ELKS1 in RIM1 condensates, maintaining the required VGCC clustering (Fig 6C–6F) for synaptic transmission. Thus, the observed immiscibility between the RIM1/RBP/VGCC

and ELKS1/liprin-α condensates may be necessary for the nano-scale organization at the active zone. Similar observations of immiscibility between the RIM1/VGCC and liprin-α3 condensates in transfected cells have also been reported recently [49]. This may ensure that highly condensed active zone proteins maintain their different localization and clustering, allowing them to form distinct functional assemblies at different sites during neurotransmitter release. Notably, a recent work by Ghelani *and colleagues* discovered that the VGCC nanocluster undergoes a compaction transition in the active zones of *Drosophila* neuromuscular junctions, and this compaction requires the involvement of the ELKS homolog, BRP [53]. Interestingly, our in vitro LLPS assay shows that the liprin-α/RIM interaction confines ELKS1 to the periphery of RIM1 condensates (Fig 6C). This spatial organization of ELKS1 results in the central localization of highly concentrated NCav_CT within RIM1 condensates (Fig 6D), a pattern resembling the VGCC compaction observed in *Drosophila*. This further highlights the potential of LLPS to play a regulatory role in the dynamic distribution of VGCCs and other proteins in the active zone.

The intricate interaction network among active zone proteins is likely to provide a critical layer of functional redundancy that ensures the robustness and adaptability of synaptic connections [1,3]. For instance, the disruption of the liprin-α/RIM interaction does not completely block the recruitment of RIM1 to the active zone (Fig 5A and 5B), presumably due to compensation by the binding of RIM to RBP and ELKS (Fig 6A). This compensatory effect explains the significant yet not profound impairment in synaptic function (Fig 5). Given the role of liprin-α in early synaptic development [11,14,22,55], the liprin-α/RIM interaction could nevertheless be critical for proper synapse assembly. The compromised RIM protein recruitment by disruptive mutations in liprin-α may still cause defects to synaptic homeostasis, contributing to pathological conditions like neurodevelopmental diseases associated with genetic variants affecting liprin-α proteins [33,34]. Importantly, this interaction network not only connects active zone core components but also integrates with other presynaptic components, like synapsin and piccolo. They ensure the rapid synaptic vesicle motility and accessibility of vesicles within presynapses, as well as facilitate the short-distance vesicle transport, through diverse LLPS-dependent mechanisms [56,57]. Considering the role of liprin-α/RIM interaction in regulating RIM LLPS, this network may orchestrate the dynamic interactions among various LLPS-mediated presynaptic condensates, directing synaptic vesicles from the reserve pool to the active zone for efficient neurotransmitter release.

Interestingly, Rabphilin-3A and synaptotagmin, known for their roles in the docking and fusion of synaptic vesicles, also utilize their C2B domains to interact with the coiled-coil structure of the SNARE complex through multiple interfaces [58,59]. The similarity between this mode of binding to that of the liprin-α/RIM complex suggests a general mechanism by which multi-interface binding can modulate the assembly and function of protein complexes in response to synaptic activity. In addition, the C2B domains of Rabphilin-3A and synaptotagmin are also involved in lipid binding [60]. The binding of lipids to the C2B domain could serve as a molecular switch that influences the conformation, localization, or interaction partners of these C2B-containing proteins (Fig 2G), which have been extensively studied in synaptotagmin [61–63]. Given the similar role of the C2B domain of RIM1 in vesicle release [29], it is likely that the coupling mechanism between membrane and liprin-α interactions of the C2B domain in RIMs also contributes to neurotransmitter release, which is compelling to further investigate. Furthermore, recent in vitro studies have demonstrated that synaptotagmin can also undergo LLPS [64,65], which may potentially impact synaptotagmin-mediated vesicle release. Considering the involvement of lipid binding in the oligomerization of the synaptotagmin C2B domain [61], how C2B dimerization and lipid binding might regulate RIMs and their LLPS-dependent processes remains an interesting question for future research.

Our study has limitations that should be considered when interpreting our findings. First, disrupting the liprin-α/RIM interaction did not significantly reduce the abundance of calcium channels at the active zone (Fig 5H), but changed the EGTA-sensitivity of evoked currents (Fig 7D). This suggests a potential impairment in the nanoscale organization of these channels within the active zone. Future studies employing cutting-edge superresolution techniques, such as STORM or single-particle PALM, would be valuable to validate our findings by assessing the nanoscale distribution of RIM, liprin-α, ELKS, RBP, and calcium channels in human neurons. Second, due to the technical challenges in manipulating

condensates in vivo, our LLPS results rely exclusively on in vitro mixing assays. Future studies that validate and extend these results in intact in vivo preparations, perhaps in more accessible model organisms such as *C. elegans*, might help assess the in vivo relevance of our findings.

## Methods

### Constructs

Human liprin-α2 (GenBank: AF034799.1) CC12 truncation with an N-terminal $His_6$-SUMO tag was generated in our previous study [23]. The CC2 (residues 259–542), CC2N (residues 300–404), and CC2C (residues 405–542) regions were subcloned into a modified pET32a vector with N-terminal thioredoxin (Trx)-$His_6$-tag and an HRV 3C protease cutting site. The full-length liprin-α2 was cloned into the pCAG vector with an N-terminal MBP tag followed by a 3C-protease cutting site. Plasmids encoding rat ELKS1 (Genbank: NM_170788.2), RIM1 (specifically the RIM1α isoform, Genbank: XM_017596673.1), and RBP2 (GenBank: XM_017598284.1) were kind gifts from Prof. Mingjie Zhang. For crystallization, RIM1_C2B (residues 1166–1334) was subcloned into a modified pET28a vector with an N-terminal $His_6$-SUMO tag. Full-length RIM1 was inserted into the pCAG vector with an N-terminal FLAG tag. The full-length ELKS1 was first cloned into a modified pETL7 vector with an N-terminal followed by a TEV-protease cutting site. Subsequently, $His_6$-MBP-GFP tagged ELKS1 was subcloned into the pCAG vector [66]. All point mutations in these constructs were created using a site-directed mutagenesis kit. Lentiviral rescue constructs were generated by subcloning PCR-amplified wildtype and mutant liprin-α2 cDNA, using Gibson assembly, to a lentiviral vector containing the ubiquitin promoter. All constructs were verified by DNA sequencing. All plasmids used in this study were summarized in Table B in S1 Text.

### Protein expression and purification

Trx or SUMO-tagged proteins were expressed in *Escherichia coli* BL21(DE3) cells. Transfected cells were cultured in LB medium at 37 °C with agitation at 200 rpm to reach an OD600 of ~0.8. After cooling to 16 °C, protein expression was induced with 500 μM IPTG and continued with overnight shaking at 16 °C and 200 rpm. Harvested cell pellets were lysed via high-pressure homogenization in a binding buffer (50 mM Tris pH 8.0, 500 mM NaCl, 5 mM imidazole) supplemented with 1 mM PMSF. The tagged proteins were purified using $Ni^{2+}$-NTA affinity chromatography with an elution buffer (50 mM Tris pH 8.0, 500 mM NaCl, and 250 mM imidazole). The eluted proteins were further purified by size-exclusion chromatography performed on a Superdex-200pg column (GE Healthcare) pre-equilibrated in TBS buffer (20 mM Tris pH 8.0, 100 mM NaCl, 1 mM EDTA, and 1 mM DTT). To prepare the RIM and liprin-α2 fragments for crystallization, the affinity tag was removed using HRV-3C or SUMO proteases at 4 °C overnight, followed by a second round of size-exclusion chromatography on a Superdex-75pg column (GE Healthcare) pre-equilibrated with TBS. Purified proteins were concentrated using Amicon Ultra centrifugal filters (Millipore) to ~10 mg/mL, aliquoted and stored at −80 °C after flash-freezing in liquid nitrogen. For fluorescence labeling, the Superdex-200pg column was equilibrated in a buffer containing 20 mM HEPES pH 8.0, 100 mM NaCl, 1 mM EDTA, and 1 mM DTT. The RBP2_$(SH3)_3$ fusion protein was prepared as previously reported [30].

Full-length RIM1, ELKS1, and liprin-α2 were expressed in HEK293F suspension cells (ThermoFisher Scientific), cultured in Freestyle 293 medium (OPM-293 CD05 Medium) at 37 °C supplied with 5% $CO_2$ and 80% humidity. When cell density reached $2.0 \times 10^6$ cells/mL, cells were transiently transfected using expression plasmids and polyethylenimine (PEI) (Yeasen Biotechnology). For transfection, ~0.5 mg plasmids were pre-mixed with 1 mg PEIs in 50 mL fresh medium for 15 min, and then the mixture was added to 500 mL of cell culture. After a 72-h culture, cells were collected at 4 °C by centrifugation at 1,000*g* for 20 min. The pellets were lysed in a buffer containing 50 mM Tris pH 7.5, 500 mM NaCl, 1 mM EDTA, 0.5% Triton X-100, and a protease inhibitor cocktail. Protein purification was performed using anti-FLAG affinity chromatography, with an elution buffer containing 100–500 μg of FLAG peptide (DYKDDDDK). Subsequent purification

steps involved size-exclusion chromatography on a Superdex 6 Increase column (GE Healthcare) using TBS with varying NaCl concentrations according to the biochemical properties of individual proteins. The protein quality was further checked using an HT7700 transmission electron microscope (HITACHI) with 100 kV voltage.

## Co-immunoprecipitation assay

Transfected HEK293T cells were lysed in ice-cold lysis buffer containing 50 mM Tris pH 7.5, 150 mM NaCl, 5% glycerol, 1% Triton X-100, 1 mM phenylmethylsulfonyl fluoride, 1% protease inhibitor cocktail (TargetMol, C001) for 0.5 h on ice and followed by centrifugation at 12,000$g$ for 15 min at 4 °C. The supernatant fraction was then incubated with anti-GFP conjugated agarose beads (Ktsm-life, ktsm1301) for 60 min at 4 °C. The beads were washed with the cell lysis buffer twice and resuspended with 20 µL SDS-PAGE loading buffer. The prepared samples were separated by 10% SDS-PAGE and transferred to polyvinylidene difluoride membranes (Millipore, IPVH00010). The membranes were sequentially blocked with 5% skim milk in buffer containing 50 mM Tris-HCl pH 7.4, 150 mM NaCl, and 0.1% Tween 20, immunoblotted with anti-GFP mouse monoclonal antibody (Transgen, HT801-01, dilution 1:3000) or anti-DYKDDDDK(anti-FLAG) mouse monoclonal antibody (Transgen, HT201-01, dilution 1:3000), probed with horseradish-peroxidase conjugated secondary antibodies (Cell Signaling, 7076s, dilution 1:10000) at room temperature and finally developed with a chemiluminescent substrate (BioRad, 107-5061). Protein bands were visualized on the Tanon-6011C Chemiluminescent Imaging System (Tanon Science and Technology).

## Isothermal titration calorimetry (ITC) assay

To quantitatively analyze protein–protein interaction, ITC experiments were conducted using a MicroCal PEAQ-ITC calorimeter (Malvern Panalytical). All proteins were prepared in an identical reaction buffer containing 20 mM Tris pH 8.0, 100 mM NaCl, and 1 mM EDTA. The protein concentration in the syringe was 400 µM for titrating into the reaction cell, where the concentration of target proteins was typically 40 µM. Experiments were carried out at a controlled temperature of 25 °C. Each titration involved injecting 3 µL of the syringe solution into the cell, followed by a 150-second equilibration period between injections. A titration curve contained a total of 13 titration points. The resulting data were analyzed using the MicroCal PEAQ-ITC Analysis software, applying a one-site binding model to determine the dissociation constant ($K_d$).

## Analytical size-exclusion chromatography (aSEC)

Analytical gel filtration chromatography was performed using an ÄKTA Pure system (GE Healthcare). The protein samples were loaded into a Superdex 200 Increase 10/300 GL column (GE Healthcare) pre-equilibrated with a buffer comprising 20 mM Tris-HCl pH 7.5, 100 mM NaCl, 1 mM EDTA, and 1 mM DTT.

## SEC coupled with multi-angle light scattering (SEC-MALS) assay

The SEC-MALS assay was conducted using a platform composed of a multi-angle light scattering (MALS) detector (miniDawn, Wyatt), a differential refractive index detector (Optilab, Wyatt) and a liquid chromatography system (AKTA pure, GE Healthcare). In each assay, a 100 µl sample (individual proteins or complexes) was injected into a Superdex 200 Increase 10/300 GL column (GE Healthcare) pre-equilibrated with TBS. Data were analyzed using ASTRA6 (Wyatt).

## Protein crystallization and structure determination

Crystals of the liprin-α2_CC2N/RIM1_C2B complex were grown using the sitting drop vapor-diffusion method. Protein samples of liprin-α2_CC2N and RIM1_C2B were mixed in a 1:1 ratio and concentrated to 21 mg/mL. This concentrated mixture (1 µL) was combined with an equal volume of reservoir buffer containing 28% v/v 2-propanol, 0.1 M BIS-TRIS pH 6.5, and 3% v/v polyethylene glycol 200 for crystallization tray setup. The crystallization was conducted at 16 °C,

and the resulting crystals were cryoprotected with 30% (v/v) glycerol. X-ray diffraction data were collected at the BL19U1 beamline of the Shanghai Synchrotron Radiation Facility. Diffraction data were processed using HKL2000 software [67]. The complex structure was solved by molecular replacement in PHASER [68] using the RIM1_C2B apo structure (PDB ID: 2Q3X) as the search model. Model building, adjustment, and refinement were carried out iteratively using COOT [69] and PHENIX [70]. The final models were validated by MolProbity [71] and statistics were summarized in Table A in S1 Text. All structure figures presented in the paper were prepared using PyMOL (https://www.pymol.org/).

## Fluorophore labeling of proteins

Fluorescent labeling dyes, including Cy3/Cy5/iFluor 488 NHS esters (ThermoFisher) and iFluor 405 NHS ester (AAT Bioquest), were dissolved in DMSO at a stock concentration of 5 mM and stored at −20 °C. Prior to labeling, proteins were concentrated at 5 mg/mL in a HEPES buffer at pH 7.5 to ensure specific N-terminal labeling. Labeling was performed by mixing the proteins with the corresponding fluorophores at a 1:1 molecular ratio and incubating at room temperature for 1 h. The reaction was quenched by adding a 200 mM Tris buffer. Unincorporated fluorescence was removed using a pre-equilibrated HiTrap desalting column (GE Healthcare) with the corresponding TBS buffer. Fluorescence labeling efficiency was assessed using a Nanodrop-2000 spectrophotometer (ThermoFisher). The labeled proteins were frozen and stored at −80 °C. For imaging, a sparse labeling approach was used, where the fluorescence-labeled proteins were mixed with an excess of corresponding unlabeled proteins in the same buffer, achieving a final molecular ratio of 1:100.

## Phase separation assays

**Sample preparation and imaging.** Prior to imaging experiments, all proteins used for imaging were centrifuged at 20,000$g$ for 10 min at 4 °C to remove any potential aggregates or precipitates. Protein concentrations and buffer conditions were specified in the corresponding figures or their legends. For imaging, samples were applied to the wells of 384-well glass bottom plates (P384-1.5H-N, Cellvis). Confocal images were captured using an A1R confocal microscope (Nikon) equipped with a 100X/NA oil objective lens. The fluorescence intensities of images were analyzed using ImageJ/Fiji software. For the phase separation assay involving RIM1 and liprin-α2_CC12, RIM1α and the CC12 fragment or its mutant were mixed at the desired concentration in a 200 μL microcentrifuge tube, then applied to the 384-well plates. Images were captured at different time points. For the phase separation analysis of ELKS1 condensates, MBP-tagged ELKS1 alone or mixed with full-length liprin-α2 were incubated with proteases for 20 min to remove the MBP tag. Following this, the samples were added to the 384-well plate, and the ELKS condensates were settled for an additional 20 min before image capture. For the phase separation assay involving both RIM1 and ELKS1 condensates, the MBP tag on the ELKS1 fusion protein was removed using TEV protease to produce GFP-ELKS1 condensates. After RIM1/RBP2_(SH3)$_3$ condensates had settled down for 10 min, ELKS1 condensates with full-length liprin-α2, liprin-α2_CC12 or liprin-α2_CC12 R346E mutant were added to the 384-well plate and mixed with RIM1α/RBP2_(SH3)$_3$ condensates. The mixtures were then allowed to settle for an additional 20 min before image capture. To detect the distribution of NCav_CT in the two-phase system, NCav_CT was mixed with ELKS1 condensates and then loaded onto the pre-formed RIM1α/RBP2_(SH3)$_3$ condensates.

**Sedimentation assay.** RIM1 was mixed with liprin-α2_CC12 at the specified concentration and incubated for 5 min. The mixture was centrifuged at 1,400 rpm for 5 min at room temperature. Following centrifugation, the supernatant was promptly isolated by pipette thoroughly, and the pellet was re-suspended with 20 μL of dilution buffer. Both supernatant and pellet samples on the SDS-PAGE gel were visualized using Coomassie blue R250 staining. The intensity of bands of interest was quantified using ImageJ/Fiji software.

**Fluorescence recovery after photobleaching (FRAP) assay.** In each FRAP experiment, 10 regions of interest (ROIs) were selected. Laser beams at 561 nm with 100% power were precisely applied to target the Cy3 fluorophore for photobleaching. Pre-bleach and post-bleach images were acquired with no delay time interval. Subsequent

time-lapse images were captured at 20-s intervals for a duration of 20 min to record fluorescence intensity recovery. These experiments were conducted using a Nikon A1R confocal microscope equipped with a 100X/NA oil lens. Fluorescence recovery was measured with ImageJ/Fiji by calculating the intensity at each time point. Data were processed by correcting for background fluorescence and normalizing the pre-bleaching intensity to 100% and the bleaching point intensity to 0%.

## Cell culture experiments

**Maintenance of human embryonic stem cells (hESCs).** hESCs of line WA09/H9 (RRID: CVCL_9773) were obtained from WiCell and maintained/cultured on Matrigel-coated (Corning #15505739) dishes using mTeSR Plus (StemCell Technologies #100-0276), which was changed every other day. hESC cells were kept in an incubator supplied with 5% $CO_2$ at 37 °C. All procedures followed The Robert Koch Institute guidelines for human ESC work.

**Maintenance of human embryonic kidney (HEK) cells.** HEK (HEK293T/17, ATCC CRL-11268) cells were used to produce all lentiviruses for this study. Cells were maintained in an incubator supplied with 5% CO2 at 37°C, using DMEM-Glutamax medium (Gibco #31966047) supplemented with 10% fetal bovine serum (FBS; Sigma #F7524). The medium was changed every other day, and cells were split after reaching near 70% confluence using Trypsin-EDTA (Gibco #15400054).

**Production of lentiviruses for neuronal differentiation, optogenetic control, and sparse visualization.** Lentiviruses were produced in HEK cells, as described elsewhere [72]. Briefly, two hours before transfection, at near 60% confluence, the medium was changed, and then HEK cells co-transfected using Linear Polyethylenimine 25,000 (PEI, Sigma Cat# 23966) with the following plasmids: pREV (3.9 μg), pRRE (8.1 μg), pVSVG (6 μg), and with the corresponding vector DNA using 12 μg per 75 cm$^2$ cell culture area. The following lentiviral vector DNA was used to produce lentiviruses for differentiation, optogenetic control, and sparse visualization: FU-M2rtTA, Tet-O-*Ngn2*-puromycin, Channel-rhodopsin oChiEF fused to tdTomato (termed here ChR-tdTomato), and soluble GFP [26]. Two hours after transfection, the medium was changed again with fresh DMEM media, and lentiviruses were harvested from the medium 40 h later. Specifically, the medium was first centrifuged at 1,500*g* for 10 min at 4 °C to eliminate dead cells and debris, and then lentiviral particles were pelleted by high-speed centrifugation (60,000*g* for 1.5 h), resuspended in MEM (Gibco #51200046) with 10 mM HEPES (100 μl per 30 ml of medium), aliquoted, and snap-frozen in liquid nitrogen.

**Production of lentiviruses for neuronal infection (rescue constructs).** All rescue experiments were performed using fresh (non-concentrated) lentiviruses, generated using the same protocol described above but with the following modifications. First, 2 h after transfection, the medium was replaced with Neurobasal supplemented with 2% B27 (Gibco #17504044), GlutaMAX (Gibco #35050061), and 10 mM HEPES (Gibco #15630080). Second, lentiviral particles were harvested from the medium 40 h after transfection with a low-speed (1,500*g* for 10 min at 4 °C) centrifugation to pellet dead cells and debris. The supernatant was then aliquoted and frozen at −80 °C.

**Generation of induced neurons.** Induced glutamatergic neurons were generated from control (Ctrl1) and liprin-α1 to α4 mutant (qKO1) ESC clones, as described in detail previously [73]. In brief, for each neuronal experiment, 250K hESCs were detached with Accutase (Gibco), plated on matrigel-coated wells in mTeSR Plus containing Rho kinase inhibitor (Y27632, Axon Medchem #1683), and transduced with concentrated lentiviruses FU-M2rtTA and Tet-O-*Ngn2*-puromycin, generated as described in the previous section. A day later (defined as DIV0), the medium was changed to N2 medium [DMEM/F12 (Gibco #11330032), 1% N2 supplement (Gibco 17502048) 1% non-essential amino acids (Gibco #11140050), laminin (200 ng/ml, Thermo Fisher #23017015), BDNF (10 ng/ml, Peprotech #450-02) and NT-3 (10 ng/ml, Peprotech #450-03) supplemented with Doxycycline (2 μg/ml, Alfa Aesar)] to induce expression of *Ngn2* and the puromycin resistance cassette. On DIV1, puromycin (1 mg/ml) was added to the medium, and after 48 h, selection cells were detached with Accutase (Gibco #A1110501) and re-plated on Matrigel-coated coverslips along with mouse glia (typically at a density of 150,000 iGluts per 24-well plate) in B27 medium [Neurobasal-A (Gibco #12349015) supplemented with B27 (Gibco #17504044), GlutaMAX (Gibco #35050061) laminin, BDNF, and NT-3]. From this time point on until

DIV10, the medium was replaced every second day, and cytosine arabinoside (ara-C; Sigma #C6645) was added to a final concentration of 2 μM to prevent glia overgrowth. Rescue lentiviral constructs (e.g., to express either WT or mutant liprin-α2 constructs that prevent interaction with RIM) were added to the medium on day 4. From DIV10, neuronal growth medium [Neurobasal-A supplemented with B27, GlutaMAX, and 5% FBS (Hyclone #SH30071.03HI)] was washed in and used for partial medium replacements every 3–4 days until analysis, typically at around 6 weeks in culture.

**Induced neurons for measuring evoked transmission.** To measure the impact of the slow calcium buffer EGTA on evoked synaptic transmission, the protocol for the generation of induced glutamatergic neurons was slightly modified, as it required the majority (80%) of cells to express ChR and a minority (20%) to express GFP for visualization for electrophysiological recordings. For this, cells from either control or qKO clones were further separated into two groups. In group #1, cells were infected with pFU-M2rtTA, pTet-O-*Ngn2*-puromycin, and lentiviruses expressing ChR-tdTomato [46,73]. In group #2, cells were infected with pFU-M2rtTA, pTet-O-*Ngn2*-puromycin, and lentiviruses to express soluble GFP. Four days later, cells from groups #1 and #2 were washed three times with PBS to remove any lentivirus trace attached to cell membranes, detached, mixed at a ratio of 80/20 (80% with ChR and 20% with GFP), re-seeded on Matrigel-coated coverslips along with mouse glia, and cultured as described above.

**Isolation of mouse glial cells.** For induced glutamatergic neurons to form mature functional synapses, we grew induced neurons on a monolayer of primary mouse glial cells. Isolation and culture of primary mouse glial cells were performed essentially as described [73]. In short, E21-P1 mouse cortices from wildtype C57BL6 mice were dissected and triturated with fire-polished pipettes and filtered through a cell strainer. Cells from two cortices were plated onto T75 flasks pre-coated with poly-L-lysine (5 mg/mL, Sigma #P1274) in DMEM supplemented with 10% FBS (Sigma). Upon reaching confluence, cells were dissociated by trypsinization and re-seeded twice to remove potential trace amounts of mouse neurons before the glia cell cultures were used for co-culture with induced neurons.

## Immunocytochemistry

Cultured neurons were fixed for 15 min at room temperature with a solution containing 4% Paraformaldehyde and 4% Sucrose in PBS, pH 7.4, then washed three times with PBS (each washing step was separated by 10 mins) and permeabilized with 0.1% TritonX-100 in PBS for 10 min at room temperature. Then, cultures were blocked for 1 hour (2% NGS, 1% BSA, 0,01% $NaN_3$ in PBS), and incubated in primary antibodies diluted in the blocking buffer overnight at 4 °C in a humidity chamber. The next day, neurons were washed (3×) with PBS and incubated with fluorescently labeled secondary antibodies (Jackson ImmunoResearch) for 1 hour at room temperature. After this, neurons were washed 3 times in PBS, followed by a wash in $ddH_2O$ and mounting in microscope slides using ProLong Gold mounting medium (Thermo Fisher Scientific). For PSD95 staining, a modified version of the protocol described above was utilized. Specifically, neurons were fixed in ice-cold Methanol fixing solution (90% methanol and 10% MES buffer: 100 mM MES pH 6.9, 1 mM EGTA, and 1 mM $MgCl_2$) at room temperature for 5 min, washed three times in PBS, and then incubated in blocking/permeabilizing solution (2% normal goat serum, 1% BSA 0.01% $NaN_3$, and 0.1% Triton X-100 in PBS) for 30 min, before proceeding with staining. The following primary antibodies and dilutions were used: MAP2 (Encor, 1:1000), pan-Synapsin (Proteogenix, 1:1000), PSD-95 (NeuroMab, 1:100 and Addgene 1:100 for SIM experiments), RIM1/2 (SySy, 1:200), Synaptophysin-1 (SySy, 1:200), Cav2.1 (SySy, 1:200). See Table C in S1 Text for details on the source of each antibody.

## Confocal imaging

For measurements of synapse density, synapse-rich areas were identified by extensive synapsin signals in close proximity to postsynaptic MAP2 profiles and were imaged using a confocal microscope (Leica, Germany) controlled by LAS X software (Leica, Germany). Specimens were sampled in frame mode at 1,024 × 1,024 pixels/frame resolution. Ten optical sections along the z-axis were taken for each sample and then compiled into a single maximal projection image for analysis. All the acquisition parameters were kept constant between conditions and experiments.

## Super-resolution STED imaging

STED imaging was conducted similarly as described previously [74]. Briefly, induced neurons were cultured on glass coverslips, and immunocytochemistry was performed as described above, except that five PBS washing steps were done after each antibody incubation and Alexa 488 anti-guinea pig (Thermo), STAR 580 nanobody anti-mouse (Abberior) and STAR 635P anti-rabbit were used as secondary antibodies. Image acquisition was conducted in a Leica SP8 Confocal/ STED 3× microscope with an oil immersion 100×1.44 numerical aperture objective and gated detectors (2–6 ns range for all three signals). Images were acquired from synapse-rich areas of 33.2 μm$^2$ sampled at ~16 nm per pixel. The signal from the 488 antibodies was acquired in confocal mode, and signals from the 580 and the 635 antibodies were acquired in STED mode sequentially (to avoid bleed trough) and using the same STED laser line (775 nm to maximize alignment). Line accumulation (4–8×) and frame averaging (3×) were applied. Images were acquired blindly to the genotype of the samples and identical settings were used for all the samples within an experiment/batch.

## Western blot

Protein samples were extracted from cultured neurons at DIV30-45, lysed in RIPA buffer (50 mM Tris pH 8.0, 150 mM NaCl, 0.1% sodium dodecyl sulfate, 0.5% sodium deoxycholate, and 1% Triton X-100) supplemented with PMSF (Thermo Fischer #36,978) and Complete proteinase inhibitor cocktail (Merch #11,873,580,001) for 20 min. Lysates were centrifuged at 20,000$g$ for 10 min at 4 °C, and supernatants containing solubilized proteins were collected. Protein samples were separated by SDS-PAGE in pre-cast TGX gels (Biorad). Transfer to a nitrocellulose membrane (Amersham) was performed in Towbin transfer buffer (25 mM Tris, 0.2 M glycine, and 20% methanol). Membranes were blocked with 5% non-fat milk (Aplichem) for 1 hour and primary antibodies were incubated overnight at 4°C. After washing the membranes three times with TBS-T (20 mM Tris pH 7.5, 137 mM NaCl, and 0.05% Tween-20), secondary antibodies were incubated in 1:1 TBS-T Odyssey Blocking (LI-COR # 927-50000) for 1 h. Membranes were imaged using an Odyssey CLx system (LI-COR), and bands were quantified by densitometry using Image Studio 5.2 software (LI-COR). The following primary antibodies (see Table C in S1 Text for details) were used: Tuj1 (BioLegend, 1:5000) and GFP (Thermo Fisher Scientific, 1:1000).

## Patch clamp electrophysiology

**General.** All electrophysiological recordings were done using an RC-27 chamber (Sutter Instruments) mounted under a BX51 upright microscope (Olympus). The microscope was equipped with DIC and fluorescent capabilities, and with a TTL-driven LED for optogenetic activation with millisecond precision. All recordings were done at 26 ± 1 °C using a dual TC344B temperature control system (Sutter Instruments), with neurons continuously perfused with oxygenated (95% O$_2$ and 5% CO$_2$) ASCF containing 125 mM NaCl, 2.5 mM KCl, 0.1 mM MgCl$_2$, 4 mM CaCl$_2$, 25 mM glucose, 1.25 mM NaH$_2$PO$_4$, 0.4 mM ascorbic acid, 3 mM myo-inositol, 2 mM Na-pyruvate, and 25 mM NaHCO3, pH 7.4. Neurons were approached and patched under DIC, using 3.0 ± 0.5 MegaOhm pipettes (WPI), pulled with a PC10 puller (Narishige, Japan). In all experiments, pipettes were loaded with a voltage clamp internal solution containing 125 mM Cs-gluconate, 20 mM KCl, 4 mM MgATP, 10 mM Na-phosphocreatine, 0.3 mM GTP, 0.5 mM EGTA, 2 mM QX314 (Hello Bio, #HB1030), and 10 mM HEPES, pH 7.2. Electrical signals were recorded using a Multiclamp 700B amplifier (Axon Instruments) controlled by Clampex 10.1 and Digidata 1440 digitizer (Molecular Devices).

**Spontaneous synaptic current recordings.** We performed whole-cell voltage-clamp recordings at ~70 mV holding potentials. Spontaneous excitatory currents were detected as downward deflections from the baseline.

**Evoked currents.** Evoked excitatory currents were triggered by 5 ms pulses of blue (488 nm) light generated via a CoolLED illumination system (pE-300) controlled by a TTL pulse, and recorded from GFP+/ChR- neurons (see above) in voltage clamp at ~70 mV holding potentials.

**Sucrose responses.** Cells were maintained at ~70 mV holding potentials (voltage clamp configuration) and stimulated with 0.5 M sucrose solution for 5 s. Sucrose solution was delivered in the vicinity of recorded cells (20–30 μm away) using a low resistance glass pipette (1.5 MegaOhms) connected to a custom pressure device (5 psi).

## Data analysis and statistics

Confocal images were handled and analyzed using LASX (Leica) or ImageJ/Fiji (v. 2.3.0/1.53f). In experiments aimed to measure the recruitment of RIM1, Piccolo, and RBP2 to presynaptic boutons in qKO neurons, the signal intensity of corresponding active zone markers was measured only inside ROIs defined by Synaptophysin. For STED image analysis, individual "side view" synapses were manually selected, and intensity profiles were obtained by drawing a rectangle of 1,200 × 200 nm centered in and perpendicular to the PSD-95 elongated signal using an ImageJ custom macro. Intensity profiles were recorded for all three signals, and the right alignment/orientation of the profiles was performed in R Studio. Intensity traces were obtained by averaging individual traces over the raw data values. Representative images in figures were linearly adjusted using bright and contrast identically across samples. Immunoblot images were handled and analyzed with Image Studio v. 5.2 (LI-COR). Analysis of voltage- and current-clamp recordings was done with Clampfit 10.1 or with custom-written macros in IgorPro 6.11. Electrophysiological/imaging experiments were done and analyzed with the experimenter blinded to the sample genotype/condition whenever possible. Summary data are shown as means ± standard errors of the mean (SEM) or means ± standard deviation (SD) as indicated in the figure legends. Statistical analysis was performed using Prism 9 (GraphPad Software). Datasets were first analyzed with the D'Agostino Pearson test to determine if the data had a normal (Gaussian) distribution. For between-group comparisons, unpaired two-tailed t-tests were used if data distribution was normal, or two-tailed Mann–Whitney tests for non-Gaussian datasets. For multiple-group comparisons, statistical significance was determined by ANOVA followed by post-hoc tests for multiple comparisons, or Kruskal–Wallis (KW) followed by Dunn's post hoc test for non-Gaussian datasets. ns, not significant; $****p < 0.0001$; $***p < 0.001$; $**p < 0.01$; $*p < 0.05$. See S1 Data for statistical details.

## Supporting information

**S1 Fig. Co-immunoprecipitation analysis of the CC2-mediated binding of liprin-α2 to ELKS1.** GFP-tagged liprin-α2 variants were co-expressed with Flag-tagged ELKS1 in HEK293T cells. The result indicates the CC2N fragment is essential for the binding of liprin-α2 to ELKS1. The data underlying this figure can be found in S1 Raw Images. (TIF)

**S2 Fig. Structural and sequence analysis of the liprin-α2/RIM1 interaction. (A)** Structural comparison of the two C2B molecules (e.g., C2B and C2B′ in Fig 2A) in the CC2N/C2B tetrameric complex. The corresponding interfaces I on CC2N are aligned well, indicating the symmetric binding of C2B to the dimeric CC2N coiled coil. **(B)** Multisequence alignment of the CC2 segment from liprin-α family proteins. Species abbreviations: 'h' for human, 'd' for *Drosophila*, and SYD2 as the *Caenorhabditis elegans* liprin-α homolog. Heptad repeats of coiled-coil structures in the CC2N sequence are annotated, with heptad labels in black and dark gray denoting residues involved in coiled-coil formation through hydrophobic and polar interactions, respectively. Polar interactions are indicated by dashed lines, while residues involved in interfaces I and II are marked by solid and open purple triangle-ups. Mutation sites associated with neurodevelopmental disorders are encircled. **(C)** Multisequence alignment of the C2B domains from RIM family proteins. The sequence of the C2A domain in human RIM1 was also included in the alignment for comparison. UNC10 is a RIM homolog in *C. elegans*. Residues involved in interfaces I and II are indicated by solid and open green triangle-ups, respectively. **(D)** Surface conservation analysis of the CC2N coiled-coil structure. Conservation scores for each residue were calculated based on the alignment in panel B. **(E)** Surface conservation analysis of the C2B structure. Conservation scores for each

residue were calculated based on the alignment in panel C. The highly conserved $PIP_2$-binding site is indicated by a circle.
(TIF)

**S3 Fig. Structure-based mutagenesis and ITC-based analysis of the liprin-α2/RIM1 interaction. (A)** Stereo view showing the molecular details of the coupling between the two interfaces of CC2N and C2B. Key salt bridges and hydrogen bonds are indicated by dashed lines. **(B)** ITC-based analyses of interface I mutation effects on the CC2N/C2B interaction. **(C)** ITC-based analyses of interface II mutation effects on the CC2N/C2B interaction. **(D)** Structural analysis of two interacting CC2N coiled coils at interface I. The salt bridge formed between R346$^{liprin-α2}$ and E1198$^{RIM1}$ is stabilized by E380$^{liprin-α2}$. **(E)** ITC-based analysis showing the mild disruptive effect of the E380R mutation in CC2N on the CC2N/C2B interaction. The data underlying panels B, C and E can be found in [S1 Data](S1 Data).
(TIF)

**S4 Fig. Analyses of multiple intermolecular interactions in the liprin-α2/RIM1 assembly. (A)** Crystal packing analysis of the liprin-α2_CC2N/RIM1_C2B complex. Different intermolecular interfaces are highlighted. **(B)** Dimerization fraction analysis of RIM1_C2B in solution using aSEC coupled with MALS. **(C)** Molecular weight analysis of liprin-α2_CC2 in solution using aSEC coupled with MALS. **(D)** The time course of condensate formation following the mixing of RIM1 and liprin-α2_CC12 or its mutants. The concentration of each protein within the mixture was 5 μM. **(E)** Sedimentation-based assay indicating the distribution of RIM1 full-length protein in the supernatant (S) and pellet (P) when mixed with increasing concentrations of liprin-α2_CC12. **(F)** Quantification of the RIM1 content of pellet fraction of samples shown in panel E. The data underlying panels B, C, E, and F can be found in [S1 Data](S1 Data) and [S1 Raw Images](S1 Raw Images).
(TIF)

**S5 Fig. Characterization of liprin-α2 mutations for rescue assays in liprin-α qKO neurons. (A)** Co-immunoprecipitation analysis of the binding of liprin-α2 variants to RIM1. Consistent with our ITC-based analyses ([S3B Fig](S3B Fig)), the interface I mutations in liprin-α2 disrupt the liprin-α2/RIM1 interaction. Results are repeated by three independent batches of experiments. **(B)** Co-immunoprecipitation analysis of the binding of liprin-α2 variants to ELKS1. The interface I mutations, especially E344R and R346E in liprin-α2, showed minimal interference with ELKS1 binding. Thus, these mutations were selected for the following rescue assays in liprin-α qKO neurons. **(C)** western blot analysis confirming comparable expression levels of liprin-α2 variants expressed by lentivirus transduction in liprin-α qKO neurons. **(D)** Quantifications of endogenous GFP fluorescence levels within nerve terminals (co-localized with Synapsin signals) in control (wild-type, no GFP expression) conditions, as well as in Liprin-α qKO cells transduced with lentiviruses expressing either liprin-α2-GFP WT or its mutants (ΔCC2N, E344R, and R346E). The data underlying this figure can be found in [S1 Data](S1 Data) and [S1 Raw Images](S1 Raw Images).
(TIF)

**S6 Fig. Morphological integrity of human synapses upon disruption of liprin-α/RIM complexes. (A)** Representative confocal micrographs showing presynaptic synapsin (green) puncta closely apposed to dendritic MAP (red) profiles under the indicated conditions. Scale bar: 5 μm. **(B)** Quantification of synapsin puncta density (left), size (middle), and intensity (right). The data reflects the minimal impact of liprin-α/RIM complex perturbations on synapse morphological integrity. The number of cells/batches analyzed for each condition is indicated on the right. **(C)** Microscopic analysis to identify synapses and measure the sub-synaptic distribution of presynaptic proteins using STED microscopy. Synapse-rich regions identified by Synapsin/PSD95 appositions were visualized at low magnification (left), followed by high magnification imaging (middle), with signal intensity quantified relative to the distance from postsynaptic PSD95 signals (right). The data underlying panels B and C can be found in [S1 Data](S1 Data).
(TIF)

**S7 Fig. In vitro LLPS analyses of active zone proteins using the CC12 fragment of liprin-α2. (A)** Confocal imaging analysis of the LLPS mixture containing ELKS1/liprin-α2_CC12 and RIM1 condensates. A magnified view of a representative droplet is displayed below, with a line analysis of fluorescence signal intensities along the indicated line. The concentration of each protein was 5 μM. **(B)** Confocal imaging analysis of RIM1/RBP2 co-condensates upon adding liprin-α_CC12 WT or the R346E mutant. A magnified view of a representative droplet, as boxed in the WT condition, is displayed below, with a line analysis of fluorescence signal intensities along the line. The concentration of each protein was 5 μM. **(C)** FRAP analysis of RIM1 condensates in the presence of RBP2_(SH3)$_3$ or liprin-α_CC12. **(D)** Plot analyses illustrating the intensity relationship between NCav_CT and RIM1 (left panel) or ELKS1 (right panel) fluorescence signals in RIM1 or ELKS1 condensates in the presence of liprin-α_CC12 WT or R346E. The data underlying this figure can be found in S1 Data.
(TIF)

**S8 Fig. In vitro LLPS analyses of active zone proteins using the full-length protein of liprin-α2. (A)** SDS-PAGE analysis of purified full-length liprin-α2. **(B)** Confocal imaging of ELKS droplets with or without liprin-α2. **(C)** Confocal imaging analysis of the LLPS mixture containing the ELKS1/Liprin-α2 co-condensate and the RIM1/RBP2_(CH3)$_3$ co-condensate. Magnified views of representative RIM1(a) and ELKS1(b) droplets were displayed below, with line analyses of fluorescence signal intensities along the indicated lines. The concentration of each protein within the mixture was 5 μM. The data underlying panels A and C can be found in S1 Data and S1 Raw Images.
(TIF)

**S1 Text.** Table A: X-ray data collection and refinement statistics. Table B: Summary of plasmids. Table C: Antibody identifiers.
(DOCX)

**S1 Data. All individual numerical values corresponding to the data displayed in** Figs 1B, 1C, 2D, 3F, 4B, 4D, 4F, 5A–5H, 6B–6F, 7C, 7D, **S3B, S3C, S3E, S4B, S4C, S4F, S5C, S5D, S6B, S6C, S7A–S7D, and S8C**.
(XLSX)

**S1 Raw Images. Raw images for blot and gel results reported in** S1, S4E, S5A–S5C, **and** S8A Figs.
(PDF)

## Acknowledgments

We thank the assistance of Southern University of Science and Technology (SUSTech) Core Research Facilities and Shanghai Synchrotron Radiation Facility (beamlines 17U1, 18U1, and 19U1). We thank Prof. Pilong Li for the pETL7 vector and S. Schoch for liprin-α2 antisera. Z.W. is an investigator of SUSTech Institute for Biological Electron Microscopy.

## Author contributions

**Conceptualization:** Fredrik H. Sterky, Claudio Acuna, Zhiyi Wei.

**Data curation:** Gaowei Jin, Joaquín Campos, Yang Liu, Berta Marcó de la Cruz.

**Formal analysis:** Gaowei Jin, Joaquín Campos, Yang Liu, Berta Marcó de la Cruz, Shujing Zhang, Mingfu Liang, Kaiyue Li, Xingqiao Xie, Fredrik H. Sterky, Claudio Acuna, Zhiyi Wei.

**Funding acquisition:** Joaquín Campos, Xingqiao Xie, Fredrik H. Sterky, Claudio Acuna, Zhiyi Wei.

**Investigation:** Gaowei Jin, Joaquín Campos, Yang Liu, Berta Marcó de la Cruz, Shujing Zhang, Mingfu Liang, Kaiyue Li, Zhiyi Wei.

**Project administration:** Zhiyi Wei.

**Resources:** Xingqiao Xie, Fredrik H. Sterky, Zhiyi Wei.

**Supervision:** Fredrik H. Sterky, Claudio Acuna, Zhiyi Wei.

**Validation:** Gaowei Jin, Joaquín Campos, Yang Liu.

**Visualization:** Gaowei Jin, Yang Liu, Fredrik H. Sterky, Claudio Acuna, Zhiyi Wei.

**Writing – original draft:** Claudio Acuna, Zhiyi Wei.

**Writing – review & editing:** Fredrik H. Sterky, Claudio Acuna, Zhiyi Wei.

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
