## [Editor Report · Decision Letter 0]

27 Aug 2024

Dear Dr Wei, 

Thank you for submitting your manuscript entitled "Liprin-α/RIM complex regulates the dynamic assembly of presynaptic active zones via liquid-liquid phase separation" for consideration as a Research Article by PLOS Biology. Please accept my sincere apologies for the long delay in getting back to you as we consulted with an academic editor about your submission. 

Your manuscript has now been evaluated by the PLOS Biology editorial staff, as well as by an academic editor with relevant expertise, and I am writing to let you know that we would like to send your submission out for external peer review.

Once your full submission is complete, your paper will undergo a series of checks in preparation for peer review. After your manuscript has passed the checks it will be sent out for review. To provide the metadata for your submission, please Login to Editorial Manager (https://www.editorialmanager.com/pbiology) within two working days, i.e. by Aug 29 2024 11:59PM.

Kind regards,

Richard

Richard Hodge, PhD

rhodge@plos.org

PLOS

---

## [Decision Letter · Decision Letter 1]

14 Oct 2024

Dear Dr Wei,

Thank you for your patience while your manuscript "Liprin-α/RIM complex regulates the dynamic assembly of presynaptic active zones via liquid-liquid phase separation" was peer-reviewed at PLOS Biology. Please accept my sincere apologies for the delays that you have experienced during the peer review process. Your manuscript has now been evaluated by the PLOS Biology editors, an Academic Editor with relevant expertise, and by four independent reviewers. 

In light of the reviews, which you will find at the end of this email, we would like to invite you to revise the work to thoroughly address the reviewers' reports.

As you will see below, the reviewers are generally positive about the significance of your findings and think that the structural aspects of the study are of high quality. However, the reviewers raise several overlapping concerns with the overall strength of the physiological data to support the claims. Specifically, Reviewer #2 notes that the experiments have been performed with the CC2 fragment of liprin and asks that key assays are repeated with full-length constructs. After discussions with the Academic Editor, we agree that these new analyses should be included in the revised version. 

Given the extent of revision needed, we cannot make a decision about publication until we have seen the revised manuscript and your response to the reviewers' comments. Your revised manuscript is likely to be sent for further evaluation by all or a subset of the reviewers.

**IMPORTANT - SUBMITTING YOUR REVISION**

*Re-submission Checklist*

*Published Peer Review*

*PLOS Data Policy*

*Blot and Gel Data Policy*

Sincerely,

Richard

Richard Hodge, PhD

rhodge@plos.org

REVIEWS:

Reviewer #1 (Stephan Sigrist, signs review): This study reveals the structure of the complex formed between the coiled-coil region of liprin-α2 and the C2B domain of RIM1, providing new insights into active zone assembly. The authors demonstrate that mutations in liprin-α, associated with neurodevelopmental disorders, block this complex's formation, underlining its importance in synapse development. Structural analysis shows a unique interaction between the coiled-coil and C2B dimers, which drives large protein assemblies and promotes RIM1 condensate formation, confirming liprin-α2's scaffolding role in early active zone protein recruitment.

Using human neurons lacking liprin-α (liprin-α qKOs), the authors found that while the liprin-α/RIM interaction is not required for synapse formation, it is essential for synaptic transmission and vesicle recruitment. Their analyses suggest that liprin-α2, through its interaction with RIM1, not only facilitates RIM1 accumulation at the active zone but also promotes VGCC (voltage-gated calcium channel) clustering. This clustering ensures efficient coupling between VGCC and vesicle priming sites, crucial for effective neurotransmitter release.

General Feedback:

The study offers valuable insights into the structure of the liprin-α/RIM complex, revealing its potential role in active zone assembly. However, certain claims in the paper should be reconsidered for a more measured approach given the in vitro context and the limitations of the physiological data.

Electrophysiology Approach:

* While the study is impressive, I am less enthusiastic about the electrophysiology approach, particularly in Figure 5. For example, the authors could have quantified the kinetics of spontaneous release events to assess if fusion kinetics were altered. Relying solely on sucrose treatment is risky when directly correlating it with synaptic features. The extent to which sucrose-triggered release accurately reflects the readily releasable pool (RRP) is debatable, as estimates may vary depending on the sucrose concentration used.

* I am not advocating for additional experiments, given the substantial amount of data already provided. However, the authors should recognize the limitations regarding synaptic function. Their assays capture only a limited set of parameters, such as the lack of access to evoked release kinetics, which could provide a more complete understanding of synaptic dynamics. Additionally, the small expression differences between constructs may have influenced the observed physiological effects, which warrants further consideration.

Statistical Soundness:

* Quote:

"Indeed, we observed that EGTA-AM more effectively blocked evoked release in liprin-α qKO neurons rescued with the R346E mutation compared to those rescued with the WT construct (Fig. 7C and D). This suggests an increased distance between calcium channels and calcium sensors in the release machinery when the liprin-α/RIM interaction is disrupted."

* The authors should address whether the observed "more effectively" blocked effect is statistically significant. It's important to provide robust statistical validation to support this conclusion, ensuring that the observed effect is not overstated without proper analysis.

* 

Liprin-α2/RIM1 Complex Formation:

* Statement: "Liprin-α2 and RIM1 form a large complex through multivalent binding."

Suggestion: The authors should soften this claim to reflect the limitations of the data. A revised phrasing like "Liprin-α2 and RIM1 might form a large complex through multivalent binding" or "Liprin-α2 and RIM1 can form a large complex through multivalent binding in vitro" would more accurately convey the conditional nature of this finding, as it has not been confirmed in vivo.

Discussion Language:

* The phrase "Our study unveils the unique assembly mechanism of the liprin-α/RIM complex, a critical yet less explored aspect," feels somewhat overstated considering the in vitro nature of the work and the physiological study's limitations. A more tempered approach would reflect the context and constraints of the findings more appropriately.

* Similarly, the phrase "Remarkably, our discovery of the liprin-α/RIM interaction's role in orchestrating the interplay of the RIM1 and ELKS1 condensates provides fresh insights into the nano-scale organization of the active zone" should be toned down. Given the in vitro context, a more measured statement would better represent the study's impact without overextending the conclusions.

Relevant Recent Work:

* The authors may want to reference recent work using single-molecule imaging in Drosophila, which demonstrates cooperation between ELKS-type BRP and the Ca²⁺ channel intracellular C-terminal in a compaction transition. Discussing this work could provide useful context and depth to their analysis of the liprin-α/RIM complex.

Reviewer #2: Jin and colleagues used structure biology tools to understand presynaptic active zone assembly. They determined the crystal structure of a fragment of the liprin-α2/RIM1 complex. They report that neurodevelopmental disease-associated mutations block the formation of the complex. Using some of these mutations, they showed that disrupting this interaction in neurons impairs synaptic transmission and reduces the readily releasable pool of synaptic vesicles. Both Liprin-a and RIM1 are important presynaptic active zone molecules. This is the first structural paper on the interaction between these two important AZ organizers. The structure part of this manuscript is very high quality and well executed. 

However, the synaptic transmission phenotype is quite subtle. They performed in vitro LLPS assays to understand the molecular functions of the liprina2-RIM1 interaction. They argue that this interaction might recruit Ca channel to active zone. However, several important technical concerns and results challenge this notion. First, the RIM1 reduction at the active zone in the absence of liprin-α2/RIM1 is quite subtle. Second, CaV concentration at AZ is normal, directly arguing against the hypothesis. Third, the LLPS droplet mixing assay is interesting but its in vivo relevance is unclear. Lastly, almost all experiments were performed only with the CC2 fragments of liprin. Given that other segments of liprin have been shown to play important roles in synaptic assembly, key results should be repeats using full length constructs. This is particularly important for the interesting results presented in Fig. 6. Specific comments:

1. In Fig. 4E, the authors used the size of the condensate as a measure for the strength of LLPS. The condensate size is a function of time. The longer one wait the bigger condensates will be. How was the 5 mins incubation time chosen? Since this is a quite artificial condition, it would be more convincing to show a time course of condensate growth instead of only 1 time point. 

2. If the liprin a2-RIM1 interaction is required for RIM1 recruitment to the active zone, why was there no difference in the data shown in Fig. S6C? this figure seems to be not consistent with Fig. 5. 

3. The RIM1 reduction shown in Fig.5 is quite moderate. If this small reduction in RIM1 level is responsible for the electrophysiological phenotypes reported in Fig.4, one would predict that heterozygous RIM1 knockout should show even stronger phenotypes. That should be tested. 

4. Fig. 6 used a large number of in vitro experiments to suggest that the CC12-RIM interaction help to recruit Ca channels to the condensate. However, in Fig. 5, the direct measurements of CaV level at presynaptic terminal showed no defects. This is an important question because of the highly artificial nature of the droplet experiments. Without in vivo validation, it calls the relevance into question.

5. Data presented in Fig. 7 suggests that there are defects in the coupling between Ca channels and release machinery. To reconcile this result with the lack of CaV defects in Fig. 5, it is possible that there are sub-active zone distribution defects. However, these difference needs to be measured. For example, if RIM, ELKS1 and RBP show differential sub active zone localization, it would be an exciting hypothesis that these "AZ" scaffolds are organized into non uniform clusters. At this point, the EGTA experiments do not directly show that. 

Reviewer #3 (Da Jia, signs review): Neurotransmitters are released by synaptic vesicle exocytosis at the active zone of a presynaptic nerve terminal. Action potential (AP) induces presynaptic membrane depolarization and subsequent opening of Ca2+ channels, and then triggers neurotransmitter release at the active zone of presynaptic terminathe. In this progress, active zone lies at the interface between the presynaptic terminal and the synaptic cleft, and transform a presynaptic action potential signal into a released neurotransmitter signal. This manuscript by Jin et al. describes the structure of the liprin-α2_CC2N/RIM1_C2B complex, and unveils the unique assembly mechanism of the liprin-α/RIM complex in the presynaptic active zone. The study is of overall high quality and interest. I only have a few suggestions that could future elevate the significance of the study.

1. In Fig.2F, please show the kd values of C2N and C2B.

2. In Fig.3F, A350liprin-α2 didn't directly involved in C2B binding, but contributes to the coiled-coil formation of CC2. Does the A350S mutant alter the LLPS property?

3. Interestingly, Jin et al purified RIM1_C2B was a monomer, while it forms a homodimer in their crystal structure. Which amino acids mediate the formation of dimeric interfaces? Do these amino acids alter the LLPS property?

Reviewer #4 (Drago Milovanovic, signs review): In the submitted manuscript, Jin and colleagues present a detailed mechanistic insight into the roles of the liprin-α/RIM-1 complex in the regulation of calcium channel dynamics and synaptic vesicle release. The manuscript spans across scales from the nanoscale crystal structure and characterization of liprin-α (CC2N region)/RIM-1 (C2B domain); to mesoscale formation of condensates showing the liprin-α/RIM-1 complexes regulate the distribution of ELKS1 and VGCC; to the functional impact assessed in the iPSC-derived human neurons lacking all liprin-α proteins (liprin-α qKO neurons). This is a carefully designed, thoroughly executed study that puts forward an exemplary role of how the liprin-α/RIM-1 interactions, while being dispensable for the assembly of the active zone, play a major role in fine-tuning the primed synaptic vesicles (SVs) and neuronal transmission. Therefore, I strongly endorse its publication with a few suggestions to address:

(i) The assembly of the active zone in liprin-α qKO neurons rescued with mutants that prevent the interactions between liprin-α/RIM-1 seems to be unaltered (Fig. S6). Yet, the evoked release in liprin-α qKO neurons rescued with either liprin-α WT or liprin-α (R346E/unable to make a complex with RIM-1) was significantly different (Fig. 7). How do authors ensure that R346E doesn't lead to misaligned or unspecialized synapses, which may lack a functional postsynapse? Is the frequency of fully mature pre-/post-synaptic assemblies the same in these different rescue experiments? Control staining with a postsynaptic marker would ensure that in rescue experiments of liprin-α qKO neurons, there is a proper apposition of the postsynaptic density with the presynaptic terminal. 

(ii) Given that liprin-α/RIM-1 complex regulates the distribution of proteins and vesicles in the active zone/RIM-1 condensates, the authors should discuss this in the context of synapsin/SV phase at the synapse, which ensures the rapid SV motility and accessibility of vesicles within presynapse (PMID: 37872159), as well as the short-distance vesicle transport proposed to channel SVs from the synapsin/SV condensates to the active zone (PMID: 38552623).

(iii) Given the relevance of C2B domains within a range of presynaptic proteins, the authors should also discuss the influence that liprin-α/RIM-1 complexes may have on the recently reported condensation of synaptotagmin-1 (PMID: 38177243 and PMID: 38980206).

---

## [Decision Letter · Decision Letter 2]

20 Mar 2025

Dear Dr Wei,

Thank you for your patience while we considered your revised manuscript "Liprin-α/RIM complex regulates the dynamic assembly of presynaptic active zones via liquid-liquid phase separation" for publication as a Research Article at PLOS Biology. Please accept my sincere apologies for the delays that you have experienced during this round of the peer review process. This revised version of your manuscript has been evaluated by the PLOS Biology editors, the Academic Editor and three of the original reviewers.

Based on the reviews, I am pleased to say that we are likely to accept this manuscript for publication, provided you satisfactorily address the following data and other policy-related requests that I have provided below (A-H):

(A) We note that the first two comments provided by Reviewer #2 may not have been directly addressed in the manuscript text. We would strongly encourage the addition of a limitations paragraph in the discussion section in instances where these comments have not been experimentally addressed.

(B) We would like to suggest a minor edit to the title as follows:

“The liprin-α/RIM complex regulates the dynamic assembly of presynaptic active zones via liquid-liquid phase separation”

(C) You may be aware of the PLOS Data Policy, which requires that all data be made available without restriction: http://journals.plos.org/plosbiology/s/data-availability. For more information, please also see this editorial: http://dx.doi.org/10.1371/journal.pbio.1001797

-Supplementary files (e.g., excel). Please ensure that all data files are uploaded as 'Supporting Information' and are invariably referred to (in the manuscript, figure legends, and the Description field when uploading your files) using the following format verbatim: S1 Data, S2 Data, etc. Multiple panels of a single or even several figures can be included as multiple sheets in one excel file that is saved using exactly the following convention: S1_Data.xlsx (using an underscore).

-Deposition in a publicly available repository. Please also provide the accession code or a reviewer link so that we may view your data before publication. 

Figure 1B-C, 2D, 3F, 4B, 4D, 4F, 5A-H, 6B-F, 7C-D, S3B-C, S3E, S4B-C, S4F, S5C-D, S6B-C, S7A-D, S8C

(D) Thank you for providing the structural data in the PDB database (8Z22). However, we note that the data is currently on hold for release. We ask that you please make the structures publicly available at this stage before publication.

(E) Please also ensure that each of the relevant figure legends in your manuscript include information on *WHERE THE UNDERLYING DATA CAN BE FOUND*, and ensure your supplemental data file/s has a legend.

(F) We require the original, uncropped and minimally adjusted images supporting all blot and gel results reported in the following Figures:

Figure S1, S4E, S5A-C, S8A

We will require these files before a manuscript can be accepted so please prepare and upload them now. Please carefully read our guidelines for how to prepare and upload this data: https://journals.plos.org/plosbiology/s/figures#loc-blot-and-gel-reporting-requirements

(G) Per journal policy, if you have generated any custom code during the course of this investigation, please make it available without restrictions. Please ensure that the code is sufficiently well documented and reusable, and that your Data Statement in the Editorial Manager submission system accurately describes where your code can be found. 

(H) Please note that per journal policy, the model system/species studied should be clearly stated in the abstract of your manuscript. 

We expect to receive your revised manuscript within two weeks. 

*Published Peer Review History*

*Press*

Best regards,

Richard

Richard Hodge, PhD

rhodge@plos.org

Reviewer remarks:

Reviewer #1 (Stephan Sigrist, signs review): I am impressed by their revision, they reacted to essentially all my questions in a satisfactory manner. I do recommend acceptance unreservedly now. 

Reviewer #3 (Da Jia, signs review): The authors have addressed all my concerns

Reviewer #4 (Dragomir Milovanovic, signs review): The authors successfully addressed all the points raised. I fully endorse the publishing of this study in PLOS Biology.

---

## [Editor Report · Decision Letter 3]

28 Mar 2025

Dear Dr Wei,

On behalf of my colleagues and the Academic Editor, Cody Smith, I am pleased to say that we can accept your manuscript for publication, provided you address any remaining formatting and reporting issues. These will be detailed in an email you should receive within 2-3 business days from our colleagues in the journal operations team; no action is required from you until then. Please note that we will not be able to formally accept your manuscript and schedule it for publication until you have completed any requested changes.

PRESS

Best wishes, 

Richard

Richard Hodge, PhD

rhodge@plos.org

PLOS
